# Surface flux estimates derived from UAS-based mole fraction measurements by means of a nocturnal boundary layer budget approach

Martin Kunz[1], Jost V. Lavric[1], Rainer Gasche[2], Christoph Gerbig[1], Richard H. Grant[3], Frank-Thomas Koch[4], Marcus Schumacher[4,5], Benjamin Wolf[2], and Matthias Zeeman[2]

[1]Max Planck Institute for Biogeochemistry, Jena, Germany
[2]Institute of Meteorology and Climate Research (IMK-IFU), Karlsruhe Institute of Technology, Garmisch-Partenkirchen, Germany
[3]Purdue University, West Lafayette, IN, USA
[4]Deutscher Wetterdienst, Meteorologisches Observatorium Hohenpeissenberg, Germany
[5]now at Alfred Wegener Institute, Helmholtz Centre for Polar and Marine Research (AWI), Neumayer station III, Antarctica

**Correspondence:** Martin Kunz (mkunz@bgc-jena.mpg.de)

**Abstract.** The carbon exchange between ecosystems and the atmosphere has a large influence on the Earth system and specifically on the climate. This exchange is therefore being studied intensively, often using the eddy covariance (EC) technique. EC measurements provide reliable results under turbulent atmospheric conditions, but under calm and stable conditions – as they often occur at night – these measurements are known to misrepresent exchange fluxes. Nocturnal boundary layer (NBL) budgets can provide independent flux estimates under stable conditions, but their application so far has been limited by rather high cost and practical difficulties. Unmanned aircraft systems (UASs) equipped with trace gas analysers have the potential to make this method more accessible. We present the methodology and results of a proof of concept study carried out during the ScaleX 2016 campaign. Successive vertical profiles of carbon dioxide dry air mole fraction in the NBL were taken with a compact analyser carried by a UAS. We estimate an average carbon dioxide flux of $12\,\mu\mathrm{mol}\cdot\mathrm{m}^{-2}\cdot\mathrm{s}^{-1}$, which is plausible for nocturnal respiration in this region in summer. Transport modelling suggests that the NBL budgets represent an area in the order of $100\,\mathrm{km}^2$.

## 1 Introduction

The terrestrial biosphere plays a major role in the carbon cycle. It has taken up approximately one quarter of the carbon emitted by human activities since 1750 (Ciais et al., 2013), but the future development of this land sink under a changing climate is uncertain. Given its importance, the biosphere–atmosphere exchange is being studied intensively. On the ecosystem level, sources and sinks of carbon dioxide are commonly quantified using the eddy covariance (EC) technique (Baldocchi, 2003). During the day, when the air is turbulently mixed, the EC technique provides reliable direct measurements of net ecosystem exchange (NEE). However, EC measurements often misrepresent nighttime fluxes (Goulden et al., 1996; Gu et al., 2005). This is related to the stable stratification that often develops close to the surface at night. Stable conditions combined with low wind

speeds violate assumptions underlying the EC technique (see Aubinet et al., 2012 for a comprehensive discussion). Despite large efforts, there is currently no generally accepted solution how to obtain reliable measurements of nighttime fluxes using the EC technique (Gu et al., 2005; Aubinet et al., 2010; Hayek et al., 2018).

Daytime NEE consists of photosynthetic uptake and release of carbon through respiration. Nighttime NEE is governed by respiration only, as photosynthesis cannot take place without light. Photosynthetic uptake and total respiration fluxes are usually of the same order of magnitude, but with opposite sign. Therefore, even slight underestimation of nocturnal respiration can result in a considerable overestimation of an ecosystem's long-term carbon uptake. Furthermore, daytime fluxes are often partitioned into photosynthetic uptake and respiration using methods that rely on the nighttime measurements (Falge et al., 2001; Reichstein et al., 2005; Lasslop et al., 2010; Wohlfahrt and Galvagno, 2017). Errors in the nocturnal fluxes might compromise this partitioning.

The nighttime problem of EC measurements calls for error quantification and potentially correction. Ideally, this would be achieved by comparison to a method that is sensitive to fluxes in the same area, but is not based on the same assumptions as the EC technique. Unfortunately, no such method is available. Constraining the error of EC measurements must therefore rely on methods that determine fluxes on smaller and larger scales.

Enclosure-based methods and biometric approaches, including plant growth assessment and stock inventories, are often employed to obtain independent estimates for NEE (Goulden et al., 1996; Wilson and Baldocchi, 2001; Campioli et al., 2016). These methods quantify the exchange of carbon on a much smaller spatial scale than EC measurements. The chambers typically used for determining soil respiration cover an area of less than one square meter, while the EC technique is sensitive to fluxes from an area of $10^4$–$10^6$ $m^2$, depending on the site and on meteorological conditions (Chen et al., 2009). Given these different scales, inhomogeneities in the ecosystem under study, such as spatial variability of soil properties (e.g., texture, carbon content, nitrogen content), soil environmental conditions (e.g., soil temperature and moisture) or plant community composition can lead to biases in the comparison.

In order to counteract these biases, larger-scale flux estimates should be obtained in addition to enclosure based or biometric measurements when constraining the error of EC measurements. Nocturnal boundary layer (NBL) budgets, first described by Choularton et al. (1995) and Denmead et al. (1996), provide such estimates. The NBL budget method makes use of the stable stratification at night, which can act as a flux-integrating enclosure. During clear nights, the emission of thermal radiation cools down the Earth's surface much faster than the air, owing to the surface's higher emissivity. An inversion layer forms, inhibiting exchange of air between the stable NBL and the neutral residual layer above (Stull, 1988). Any tracer emitted from the surface into the atmosphere is therefore accumulated within the NBL. By measuring the rate of accumulation, the tracer flux can be estimated.

Different setups have been used for NBL budgeting. Acevedo et al. (2004) measured $CO_2$ dry air mole fractions at a 12 m tower, sampling only the lowest parts of the NBL. Although they did not sample the whole layer, they were able to determine a budget by identifying an effective accumulation height from either heat flux or balloon-borne humidity and temperature measurements and assuming a uniform accumulation rate of $CO_2$ up to this height. Winderlich et al. (2014) used $CO_2$ and $CH_4$ dry air mole fraction measurements at 6 heights on a 301 m tall tower, yielding profiles that encompass the whole NBL

during most nights. Often the NBL budget method is applied without a tower. A tethered balloon can lift a 100–300 m long hose through which a ground-based gas analyser samples air from different heights (Choularton et al., 1995; Denmead et al., 1996; Culf et al., 1999). Alternatively, a light analyser can be carried by the tethered balloon directly (Pattey et al., 2002).

Despite providing unique information, the NBL budget method has been applied only infrequently in recent years. This might be related to the cost and operational limits of towers and tethered balloons. Unmanned aircraft systems (UASs) could make the NBL budget method more accessible. UASs with payload capacities on the order of 1 kg are now available for few thousand Euros. When equipped with lightweight trace gas analysers and meteorological sensors (Kunz et al., 2018) they have the potential to probe the NBL with large flexibility at low cost. Multicopters are a particularly attractive type of UAS for this kind of study, because their vertical take-off, vertical landing and hovering capability makes them easy to operate in a range of environments.

However, the air movement caused by a UAS can disturb the NBL and thereby compromise the measurements. A reliable NBL budget can be determined only if this issue is addressed. A second challenge is not specific to UASs, but common to all NBL budgets: the area contributing to the budget depends heavily on the meteorological conditions and can extend far from the point of measurement. For a given time and site this footprint cannot be influenced by experimental design. Nevertheless, knowledge of the footprint is beneficial for the data analysis and interpretation of the results. In earlier NBL studies, this topic has received only basic treatment (Culf et al., 1999) or was ignored altogether.

To assess the suitability of UASs as measurement platforms for the creation of NBL budgets we carried out a proof of concept study. We deployed a carbon dioxide analyser on a multicopter and repeatedly sampled vertical profiles of the NBL during two nights in July 2016 as part of the ScaleX 2016 campaign in Fendt, Germany. Section 2 is a description of the site and the available ground-based instrumentation, the airborne measurement system and the unmanned aircraft. In Section 3 we explain how we dealt with the disturbance caused by the UAS, which post-processing steps we carried out and how we determined the NBL budget. Furthermore, we delineate how a Lagrangian transport model can be applied to identify the areas that contributed to the budget, i.e. how to determine the footprint of our flux estimates. In Section 4 we present and discuss the profiles taken by the UAS, the fluxes obtained from the NBL budget and a summary of the footprint analysis. We compare our observations to references and assess the robustness of our flux calculation. In Section 5 we summarise the merits and experimental challenges of our approach.

## 2   Site and instrumentation

### 2.1   Fendt site

The Fendt site is located in southern Germany in the Alpine Foreland (Fig. 1) at 11.060° N, 47.833° E (WGS84), 600 m above mean sea level. The site lies in a flat valley bordered by a gentle slope to the east and a steep slope leading to a 100 m higher plateau to the west. The valley floor is dominated by pasture and some crops, predominantly maize, which in Germany is typically sown in April or May and harvested between September and November. The slopes to the east and west are covered

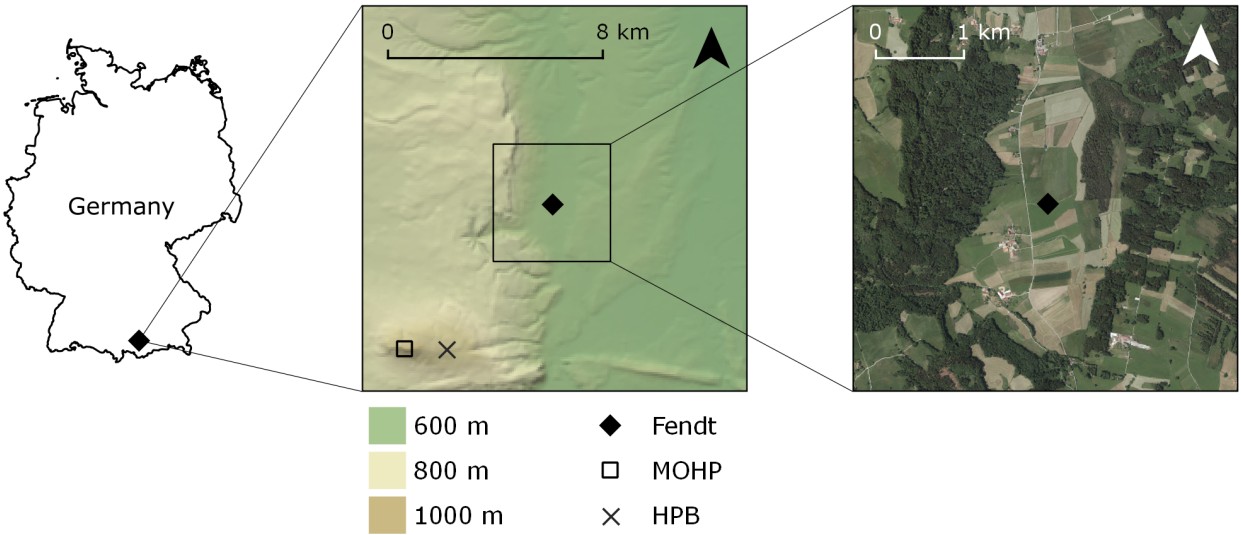

**Figure 1.** Location, topography and aerial image of the Fendt site and its surroundings. MOHP is the Meteorological Observatory Hohenpeißenberg, HPB is the ICOS station Hohenpeißenberg. Digital elevation model and aerial imagery by Bayerische Vermessungsverwaltung, www.geodaten.bayern.de.

with coniferous and mixed forest, respectively. Fendt belongs to the district Weilheim–Schongau, which has a population density of 139 km$^{-2}$ (Statistisches Bundesamt (Destatis), 2018, 35th percentile of all districts in Germany).

While soil identification at the Fendt site resulted in Stagnosols at three locations, soil organic carbon (SOC) content was determined additionally at 20 locations within a regular grid covering an area of 300 m by 300 m. SOC content in 5cm depth

varied between 4 and 11%, while at 50cm depth, values of up to 23% were obtained. The highest SOC contents were observed at the eastern side of the regular grid where a peat area is located. According to BGR (2013), organically rich soils (Cambisols and Histosols) prevail within 20 km radius around the Fendt site (Fig. 2a). The dominant land cover in this region are crops, pasture and forest (Fig. 2b).

Five kilometres south-west of the Fendt site lies an isolated, 988 m high mountain, the "Hoher Peißenberg". Close to its sum-

mit the German Weather Service (Deutscher Wetterdienst DWD) operates the Meteorological Observatory Hohenpeißenberg (MOHP) and the ICOS (Integrated Carbon Observation System) station Hohenpeißenberg (HPB, Fig. 1).

### 2.2  Ground-based instrumentation

Fendt is part of the TERENO (Terrestrial Environmental Observatories) network and is extensively instrumented for the purpose of long-term monitoring of land–atmosphere exchange (Mauder et al., 2013; Kiese et al., 2018). Complementary obser-

vations were made during the ScaleX 2016 campaign (Wolf et al., 2017). In the following, we list only those instruments that produced the data presented in this publication.

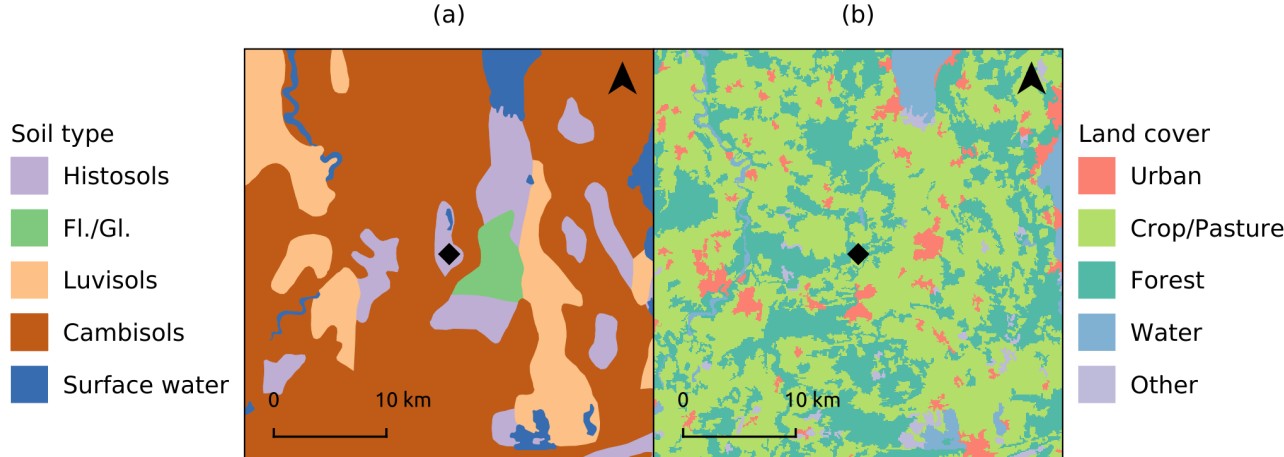

**Figure 2.** (a) Soil types in the region around the Fendt site, based on BGR (2013), denoted in WRB classification (IUSS Working Group WRB, 2015). 'Fl./Gl.' stands for 'Fluvisols/Gleysols'. (b) Simplified land cover map (CORINE 2012 v18.5, European Environment Agency, EEA (2016)) of the same region. In both panels the location of the Fendt site is marked with a black diamond.

During the ScaleX 2016 campaign, $CO_2$ dry air mole fraction at at a height of 1, 3 and 9 m above ground level was measured with a cavity ring-down spectrometer by successive sampling of air through three inlets installed at a 9 m high mast. Each inlet was sampled once every 7.5 min, with occasional interruptions due to calibrations and other measurements. An EC system installed at 3.5 m height (Zeeman et al., 2017) quantified the turbulent exchange of $CO_2$. Air temperature and upward as well as downward radiation were measured at 2 m height. Two sets of automated chambers were operated to determine the total NEE or respiration flux of grass and soil. One set comprised four LI-8100 long-term chambers (LI-COR, Lincoln, NE, USA), two with clear enclosure for measuring NEE and two with opaque enclosure for measuring respiration (Zhao et al., 2018). All four chambers covered an area of 317.8 $cm^2$ and will be referred to as "small chambers" from here on. The other set consisted of 5 custom-built opaque chambers covering an area of 2500 $cm^2$, referred to as "big chambers" hereafter. All the instruments mentioned so far were located close to each other and our UAS flew within 200 m horizontal distance to each of them.

Besides the on-site instruments we use two more data sources for our analysis. One is the observation of cloudiness at the Meteorological Observatory Hohenpeißenberg, recorded every hour either by a person or an automated instrument. We consider these 5 km distant measurements representative for Fendt, with a potential time lag on the order of 1 h in case of synoptic events. The second non-local data source is the greenhouse gas monitoring system at the ICOS station Hohenpeißenberg, situated at 934 m above mean sea level. We use its measurements of the $CO_2$ dry air mole fraction at 131 m height above ground level, i.e. at 460 m above the Fendt site.

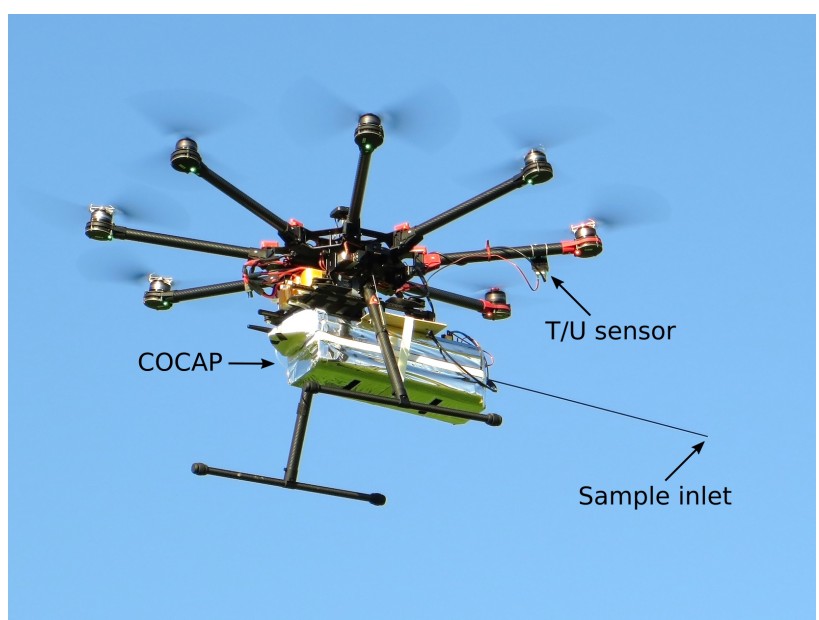

**Figure 3.** COCAP was carried by a multicopter during the ScaleX 2016 campaign. The position of the sample inlet for the $CO_2$ measurement and of the temperature and humidity sensor board are indicated.

**Table 1.** Measurement principles, uncertainties and calibration range of the airborne sensors

|  | $CO_2$ dry air mole fraction | Temperature | Pressure | Humidity |
| --- | --- | --- | --- | --- |
| Principle | nondispersive infrared | platinum resistance | piezoresistive | capacitive polymer |
| Uncertainty | see Section 4.1 | 0.15 °C | 1.5 hPa | 2 % |
| Cal. range | 380–600 $\mu mol \cdot \mu mol^{-1}$ | 0–30 °C | 400–1000 hPa | 20–90 % |

## 2.3 Airborne payload

For the study presented here, temperature, pressure, relative humidity and $CO_2$ dry air mole fraction of ambient air were measured using COCAP, the COmpact Carbon dioxide analyser for Airborne Platforms, developed at the Max Planck Institute for Biogeochemistry in Jena (see Kunz et al., 2018 for a detailed description). COCAP was mounted below the multicopter. Air
5  samples for the measurement of carbon dioxide dry air mole fraction were drawn from an inlet placed 30 cm below and 20 cm to the side of the rotors (Fig. 3). The temperature and humidity sensor board, requiring strong ventilation for fastest response, was placed directly below one of the rotors. The sensor for ambient pressure was located inside COCAP's housing, which was not hermetically sealed and therefore in equilibrium with ambient pressure.

The measurement principles employed by the different sensors as well as their measurement uncertainties are listed in
10  Table 1. The uncertainty of the calibration is included in the measurement uncertainties reported.

## 2.4 Unmanned aircraft

During ScaleX 2016 COCAP was deployed on an S1000 multicopter (SZ DJI Technology, China) controlled by a Pixhawk autopilot (3D Robotics, Berkeley, CA, USA) running the Ardupilot APM:Copter V.3.3.3 Firmware. Take-off mass of the whole system was 8 kg. The multicopter was powered by three lithium polymer batteries with a voltage of 22.2 V and a capacity of 5000 mAh each, achieving a maximum flight time of 12 min. Our special flight permit included nighttime flights, but because the take-off mass of our UAS exceeded 5 kg, all flights were limited to a maximum height of 150 m.

## 3 Methods

### 3.1 Disturbance by the UAS

A multicopter as a rotary-wing aircraft counterbalances gravity by accelerating air downwards through the movement of its rotors. The resulting displacement of air can interfere with in situ measurements, because air might be sampled at a location where it would normally not reside. In addition, volumes of air originating from different locations can be mixed together. The greater the displacement and mixing caused by the UAS, the greater is the potential impact on e.g. the measurement of a gradient. Air movement below and above the rotors is not symmetric: below a rotor, air is pushed downwards as a directed stream with high speed. In contrast, the air flow towards the rotor comes from different directions and has lower speed. The reader can easily confirm this with a fan or a hair dryer: while the outflow of air can be felt meters away, the inflow is hard to sense even near the rotor.

In view of the asymmetric flow pattern we expect that during ascent of the UAS air parcels are measured with negligible displacement from their undisturbed location. During descent, however, the sensors are moved into a volume that potentially has been flushed with air originating from several meters above. During hovering at a fixed location or during purely horizontal movement, the sensors might reside in a partially closed flow loop that extends below and aside the multicopter, effectively measuring a mixture of air from different locations.

For the study presented here, flying near the ground can have a particularly strong influence on the measurements for three reasons. Firstly, downward motion of the air stops at the ground and displaced air must move laterally or upwards, making a fast flow path back to the UAS more likely. Secondly, in our nighttime experiments the air near the ground is stably stratified. Therefore, air pushed downwards by the rotors experiences a restoring upward force, increasing the chance that closed flow loops form. Thirdly, the strongest gradients in temperature and $CO_2$ dry air mole fraction are present close to the ground, hence even a small displacement of air can have a large effect on the measured values.

In case of considerable horizontal air speed, due to either wind or horizontal flight, the rotor-induced airflow should have a smaller effect on measurements because the sampling system is moving away from air that has been displaced. We investigated this effect by flying horizontally at different speeds over a homogeneous meadow (see Section 4.4).

Based on the considerations above and the data presented in Sections 4.3 and 4.4 we determine the NBL budget only from those measurements that were taken during ascent of the multicopter. The sensitivity of the NBL-derived fluxes to inclusion of

hover and descent data is discussed in Section 4.5. Furthermore we discard COCAP's $x_{CO2}$ data collected below 9 m height for the calculation of the NBL budget. Instead, the lowest part of the $x_{CO2}$ profile is defined by the stationary measurements at the 9 m mast at 1, 3 and 9 m height. Pressure and temperature at these levels are interpolated from COCAP's measurements. During flight, the horizontal distance between COCAP and the 9 m mast was less than 150 m at any time. Hence, we do not expect pronounced horizontal gradients in $x_{CO2}$ between the measurement locations. In Sect. 4.6 we discuss how the NBL-derived fluxes are affected if the data from the 9 m mast is not used.

## 3.2 Correction for response time of sensors

On a moving platform the finite response time of sensors can be a source of measurement error, as the response time distorts the attribution of data points to time and location. COCAP's pressure and temperature sensors are fast enough for this effect to be neglected, but both the humidity and the $CO_2$ sensor require correction.

The response of a capacitive humidity sensor can be expressed as Miloshevich et al. (2004)

$$\frac{dU_m}{dt} = k(U_a - U_m).\tag{1}$$

$U_a$ and $U_m$ are ambient and measured relative humidity, respectively. The coefficient $k$ is inversely related to the sensor's response time and might be temperature-dependent. Solving Equation 1 for $U_a$ provides a simple way to compute true humidity from measurements. We use a 4th order Savitzky–Golay filter (Savitzky and Golay, 1964) with a length of 15 samples to compute $\frac{dU_m}{dt}$ while keeping high-frequency noise at an acceptable level. The coefficient $k = 14\,\mathrm{s}$ was determined by an optimisation that minimises the difference between the corrected humidity profiles for ascent and descent. We tested a linear and quadratic dependence of $k$ on ambient temperature, but found no improvement that would justify the additional degrees of freedom in the model.

The response of COCAP's $CO_2$ sensor is more complex. Its response to step-changes in $CO_2$ dry air mole fraction can be approximated as

$$x_{SC}(t) = \begin{cases} x_0 & \text{if } t < t_d \\ a(x_0 - x_\infty)e^{(t_d - t)/\tau_1} + (1-a)(x_0 - x_\infty)e^{(t_d - t)/\tau_2} + x_\infty & \text{if } t \geq t_d. \end{cases}\tag{2}$$

Here $x_0$ and $x_\infty$ denote the $CO_2$ dry air mole fraction before and infinitely long after the step change, respectively, and $t_d$ is the sensor's dead time.

We determined the coefficient $a$, the dead time $t_d$ as well as the time constants $\tau_1$ and $\tau_2$ from experimental data collected in the field. With COCAP running in flight configuration, i.e. with the inlet tube attached, we connected a tube with gas flowing from a cylinder. We observed a dead time of $t_d = 5\,s$ between making the connection and the first change of COCAP's reading. The remaining parameters were found by least-squares regression of Equation 2 to the data, yielding $\tau_1 = 27\,\mathrm{s}$ and $\tau_2 = 3.2\,\mathrm{s}$.

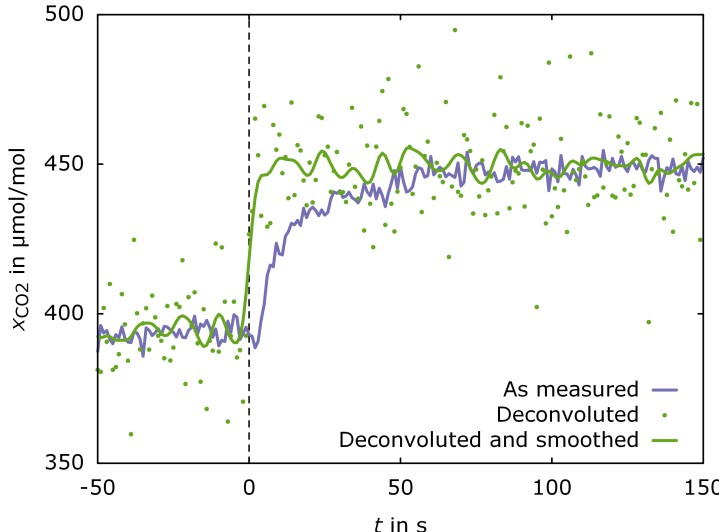

**Figure 4.** Response of COCAP to an abrupt change in $CO_2$ dry air mole fraction $x_{CO2}$ at the inlet at time $t = 0\,$s. The measured signal reveals a dead time of 5 s of the sampling system. Furthermore, the step change in $x_{CO2}$ is smoothed out. Both effects are removed by deconvolution at the cost of higher noise. Smoothing the deconvoluted signal reduces the noise with only minor impact on the time response. Smoothing was carried out by convolution with a Gaussian function of 5 s full width at half maximum (FWHM).

Ignoring noise and calibration error, any $CO_2$ signal $x_a$ is reported by COCAP as the convolution of $x_a$ with the $CO_2$ sensor's instrument function $f$ (see Kunz et al. (2018)):

$$x_m(t) = (x_a * f)(t) \tag{3}$$

$$\text{with } f = \begin{cases} 0 & \text{if } t < t_d \\ \frac{1}{x_\infty - x_0} \cdot \frac{dx_{SC}}{dt} & \text{if } t \geq t_d. \end{cases} \tag{4}$$

As $x_{SC}$ is known from experiment, $f$ can be calculated. The ambient signal $x_a$ can be recovered from the measured signal $x_m$ by deconvolution (Fig. 4). We carried out the deconvolution in Fourier space where it is equal to a division. In the numerical implementation it is important to discretise $f$ in a way that does not underestimate the slope of $f$ between the time steps $t_d$ and $t_d + \Delta t$, because doing so would lead to a strong enhancement of the noise during the deconvolution of $x_m$ with $f$. The opposite error, i.e. overestimating the slope between $t_d$ and $t_d + \Delta t$, is less critical and just results in a slight smoothing.

### 3.3 Calculation of the NBL budget

For a parcel of air in the atmosphere the following continuity equation holds (Leuning, 2004, Equation 6.2):

$$\sigma = \frac{\partial c}{\partial t} + \nabla \cdot (c\overrightarrow{u}) \tag{5}$$

$$= \frac{\partial c}{\partial t} + c(\nabla \cdot \overrightarrow{u}) + u\frac{\partial c}{\partial x} + v\frac{\partial c}{\partial y} + w\frac{\partial c}{\partial z}. \tag{6}$$

Here $\sigma$ is the strength of a volume source (or sink) of carbon dioxide (in units $\mu\text{mol} \cdot \text{m}^{-3} \cdot \text{s}^{-1}$), $c$ is the concentration of carbon dioxide and $t$ denotes time. The components $u$, $v$ and $w$ of the wind vector $\overrightarrow{u}$ point towards east ($x$-direction[1]), towards north ($y$-direction) and upwards ($z$-direction), respectively. Molecular diffusion is neglected. Due to continuity of the air flow, the term $c(\nabla \cdot \overrightarrow{u})$ equals zero. If we follow an air parcel as it is transported by horizontal winds, those terms that contain a horizontal wind component vanish as well and Equation 6 is reduced to

$$\sigma = \frac{\partial c}{\partial t} + w\frac{\partial c}{\partial z}. \tag{7}$$

Now we integrate vertically over those air parcels that form a vertical column over our site at the time of measurement (at earlier or later points in time, the air parcels are not aligned in a vertical column, unless the wind vector is equal at all heights):

$$\int_0^{z_t} \sigma\,dz = S = \int_0^{z_t} \frac{\partial c}{\partial t}\,dz + \int_0^{z_t} w\frac{\partial c}{\partial z}\,dz. \tag{8}$$

For our measurements, we choose $z_t = 125\,\text{m}$, so all biotic sources of carbon dioxide are within the column and $S$ represents NEE.

Between a reference time $t_0$ (see below) and the time of a flight $t_F$ the column has accumulated

$$\int_{t_0}^{t_F} S\,dt = \int_{t_0}^{t_F}\int_0^{z_t} \frac{\partial c}{\partial t}\,dz\,dt + \int_{t_0}^{t_F}\int_0^{z_t} w\frac{\partial c}{\partial z}\,dz\,dt \tag{9}$$

$$= \int_0^{z_t} c(z,t_F)\,dz - \int_0^{z_t} c(z,t_0)\,dz + \int_{t_0}^{t_F}\int_0^{z_t} w\frac{\partial c}{\partial z}\,dz\,dt. \tag{10}$$

---

[1]Only here and in Eq. 6 $x$ denotes a coordinate in space. Elsewhere in this publication $x$ denotes the dry air mole fraction of a substance.

Dividing the accumulated amount of $CO_2$ by $t_F - t_0$ yields NEE averaged over this time span, denoted $\overline{S}$:

$$\frac{\int_{t_0}^{t_F} S\,dt}{t_F - t_0} \;=\; \overline{S} \;=\; \frac{\int_0^{z_t} c(z,t_F)\,dz - \int_0^{z_t} c(z,t_0)\,dz}{t_F - t_0} \;+\; \frac{\int_{t_0}^{t_F}\int_0^{z_t} w\frac{\partial c}{\partial z}\,dz\,dt}{t_F - t_0}. \tag{11}$$

$$\underbrace{\phantom{XXXXXXXXXXX}}_{\text{(A)}} \qquad\qquad \underbrace{\phantom{XXXXXXXX}}_{\text{(B)}}$$

Term A represents the enhancement in $CO_2$ concentration and term B the vertical exchange of $CO_2$. We choose $t_0$ as the time when the surface radiation balance becomes negative, i.e. the time when the stable NBL starts to form. A positive $\overline{S}$ means emission of $CO_2$ from the surface into the atmosphere.

The $CO_2$ concentration $c$ can be calculated from $CO_2$ dry air mole fraction $x_{CO2}$, air temperature $T$ and dry pressure $p_d$ using the ideal gas constant $R$ (Foken et al., 2012, p. 5):

$$c = \frac{p_d}{RT} x_{CO2}. \tag{12}$$

As COCAP measures $x_{CO2}$, $T$, $p$ and relative humidity $U$, the integral $\int_0^{z_t} c(z,t_F)\,dz$ in Equation 11 term A is readily computed. However, each air parcel is sampled only once, at the time $t_F$ when it passes the Fendt site. In order to evaluate the second integral in term A, $\int_0^{z_t} c(z,t_0)\,dz$, we assume horizontal and vertical homogeneity of the $CO_2$ dry air mole fraction at the time
$t_0$, i.e. $x_{CO2}(t_0)$ is assumed to be constant within the spatial domain relevant for our experiments. Thus we can calculate the second integral from the measurement of a different column at Fendt at $t_0$.

Note that this is a weaker assumption than the horizontal homogeneity of the $CO_2$ concentration *in the NBL* presumed in other studies Choularton et al. (1995); Culf et al. (1999); Acevedo et al. (2004). All natural environments exhibit a certain horizontal heterogeneity in $S$. Daytime and nighttime $CO_2$ flux of a vegetated area are usually of the same order of magnitude,
although different in sign. Before $t_0$, the convective boundary layer is well mixed up to a height of typically 1 km, whereas after $t_0$ a strong NBL confines emissions from the surface to the lowest $\approx 100\,\mathrm{m}$ of the atmosphere (see Sect. 4.3). Therefore, the horizontal heterogeneity in $c$ caused by the horizontal heterogeneity in $S$ is one order of magnitude smaller during day time than during the night. The convective mixing during the day also keeps vertical gradients inside the boundary layer low, hence the approximation of $x_{CO2}(t_0)$ being independent of $z$ is justified.
Term B in Equation 11 contains the product $w\,\partial c/\partial z$, which generally includes both turbulent exchange and subsidence. However, when a stable NBL has developed, little turbulent exchange takes place across the top of the NBL. In the statically neutral residual layer above the NBL, turbulence is present, but the vertical concentration gradient in the residual layer and as a consequence the net vertical transport of $CO_2$ is small. Hence, we neglect turbulent exchange and identify $w\,\partial c/\partial z$ with subsidence or lifting. The vertical wind speed $w$ due to subsidence at a height of 100 m is usually on the order of $100\,\mathrm{m{\cdot}d^{-1}}$,

i.e. very low and therefore challenging to measure. We retrieve an estimate of $w$ from the Integrated Forecast System (IFS) run by the European Centre for Medium-Range Weather Forecasts (ECMWF).

In order to calculate $\partial c / \partial z$ at different times between $t_0$ and $t_F$, we use a simple model for the growth of the NBL:

$$x_{\mathrm{CO2}}(t,z) = x_{\mathrm{CO2}}(t_F, \frac{t_F - t_0}{t - t_0} z) \qquad (t_0 < t \leq t_F). \tag{13}$$

Where $\frac{t_F - t_0}{t - t_0} z$ exceeds the maximum height of the profile measured at time $t_F$ we assume the $CO_2$ dry air mole fraction to be equal to $x_{\mathrm{CO2}}(t_0)$. This model for the growth of the NBL can be visualized best by starting at $t_F$ and looking back in time. At $t = t_\mathrm{F}$ the factor $(t_F - t_0)/(t - t_F)$ is equal to unity and the model yields the measured profile. At earlier times, the measured profile is compressed in $z$-direction, such that the height of the NBL decreases linearly as we go back in time. As $t$ approaches

$t_0$, the model yields a thin layer enriched with $CO_2$ at the surface and a constant $CO_2$ dry air mole fraction of $x_{\mathrm{CO2}}(t_0)$ above.

The concentration $c(t,z)$ is calculated from $x_{\mathrm{CO2}}(t,z)$ using Equation 12. To this end, we determine $p_d(t,z)$ and $T(t,z)$ by linear interpolation in time between the first profile of the night and the profile measured at $t_F$.

In summary, the model for the growth of the NBL represents four simplifying assumptions: (1) During the night, the NBL height increases linearly, (2) the integral $\int_0^{z_t} x_{\mathrm{CO2}}(t,z)\,dz$ increases linearly with time, (3) the shape of the $x_{\mathrm{CO2}}$ profile within

the NBL remains the same throughout the night and (4) the dry pressure and temperature of an air column measured at Fendt are representative for the whole footprint of the measurement (see Sect. 3.4).

### 3.4   Footprint calculation

The columns of air probed at Fendt at different times had a different history, depending on the wind field and atmospheric stability. Atmospheric transport models can identify the surface areas that have contributed to an observed tracer concentration,

i.e. the footprint of an observation. We simulate atmospheric transport with STILT, the Stochastic Time-Inverted Lagrangian Transport model (Lin et al., 2003; Gerbig et al., 2003a), which is based on NOAA's HYSPLIT particle dispersion model (Stein et al., 2015). In our configuration, STILT launches 10 000 air parcels at different heights (see below) at every full hour during the period of our NBL measurements. Driven by meteorological data with a resolution of $0.1° \times 0.1°$ from the ECMWF IFS (European Centre for Medium-Range Weather Forecasts Integrated Forecast System), STILT calculates the back trajectories

of these parcels until 10 h in the past. For each time step of the simulated transport the model determines the sensitivity of the $CO_2$ concentration in the parcel to the $CO_2$ flux at the surface. To do so, the height up to which mixing occurs is estimated from the meteorological data using a modified Richardson number method (Lin et al., 2003). Surface fluxes influence air parcels within a column that extends from the surface to 1/2 this height in each time step (Gerbig et al., 2003b).

The back trajectories calculated by STILT are then aggregated into mole fraction footprints on a regular grid with a resolution

of 2 km×2 km. As explained in the previous section we assume the $x_{\mathrm{CO2}}$ distribution to be homogeneous in the lateral and horizontal directions at time $t_0$. We therefore restrict the aggregation to that part of each back trajectory that lies between $t_0$ and the time of measurement.

A single STILT run determines the sensitivity of an observation at a specific height to upwind fluxes. Formally, the mole fraction footprint of a measurement taken at the geographic location $(l_1, l_2)$ at time $t$ and observation height $z$ can be written as $f(l_1, l_2, z, t \,|\, l_{G1}, l_{G2})$. As all our measurements were taken at the same horizontal location, the dependency of $f$ on $l_1$ and $l_2$ will be omitted hereafter. The mole fraction footprint is a function whose value is the sensitivity to the surface flux at the grid cell specified by $(l_{G1}, l_{G2})$ in units of $[f] = \mu\text{mol} \cdot \text{mol}^{-1} \cdot \mu\text{mol}^{-1} \cdot \text{m}^2 \cdot \text{s}$. To determine the relative contribution of surface fluxes in different areas to our NBL-derived fluxes we need a different, but related function, the flux footprint $f_F$ with units $[f_F] = \mu\text{mol} \cdot \text{m}^{-2} \cdot \text{s}^{-1} \cdot \mu\text{mol}^{-1} \cdot \text{m}^2 \cdot \text{s} = 1$. The flux footprint is calculated by integration over an array of mole fraction footprints for different measurement heights, i.e. analogous to Equation 11 term A and Equation 12:

$$f_F(t \,|\, l_{G1}, l_{G2}) = \frac{\displaystyle\int_0^{z_t} \frac{p_d(z, t)}{RT(z, t)} \cdot f(z, t \,|\, l_{G1}, l_{G2}) dz}{t - t_0} \tag{14}$$

Dry pressure $p_d$ and air temperature $T$ at time $t$ and height $z$ are inter- or extrapolated from the measured profiles. The ensemble of mole fraction footprints comprises footprints for 12 different measurement heights between 10 m and 120 m in 10 m-steps.

The meteorological data we use have a horizontal resolution of $0.1° \times 0.1°$, corresponding to 11 km×8 km at the latitude of Fendt. Terrain features that are smaller than a grid cell, like the valley slope to the west of the Fendt site, cannot be represented at these resolutions. The vertical resolution of the meteorological data depends on height above ground. The lowest layer extends from the ground to 10 m height, the following 5 layers extend from the top of the previous layer to 31 m, 55 m, 80 m, 108 m and 138 m, respectively. The temporal resolution of the ECMWF IFS data is 3 h.

## 4 Results and discussion

### 4.1 Uncertainty of $x_{CO2}$ measurements

The uncertainty of COCAP's $x_{CO2}$ measurements due to drift and calibration errors is about $1 \, \mu\text{mol} \cdot \text{mol}^{-1}$ (Kunz et al., 2018). The additional uncertainty caused by noise is dependent on the data treatment, as can be seen from Fig. 5. This Allan deviation plot (Allan, 1987) is based on measurements of a gas standard ($x_{CO2} = 447.44 \, \mu\text{mol} \cdot \text{mol}^{-1}$) over a period of 1.4 h, taken in the field on 6 July 2016.

The curves illustrate that deconvolution amplifies noise in the data by a factor of 7 if no averaging is applied ($\tau = 1$ s). However, if more than 100 samples are averaged ($\tau \geq 100$ s), the difference between original and deconvoluted data becomes negligible and the uncertainty of the average due to noise is lower than $0.5 \, \mu\text{mol} \cdot \text{mol}^{-1}$. All our column integrals (see Section 3.3) have a sample size larger than 100.

Figure 5 also shows that the Allan deviation of deconvoluted data that has been smoothed by convolution with a Gaussian function of 10 s full width at half maximum (FWHM) increases between $\tau = 1$ s and $\tau = 5$ s. This increase is an artefact caused by the autocorrelation that the smoothing induces. If COCAP was perfectly calibrated and exhibited no drift, any single

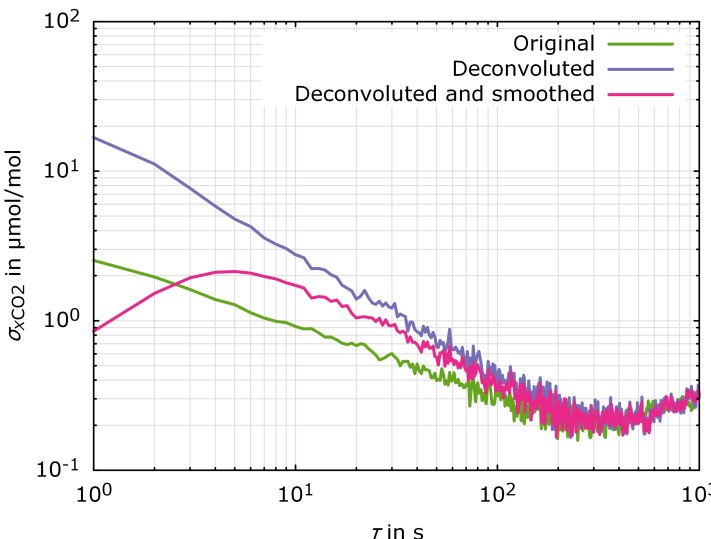

**Figure 5.** Allan deviation of $CO_2$ dry air mole fraction $\sigma_{xCO2}$ as measured, after deconvolution and after deconvolution followed by smoothing (convolution with a Gaussian function of 5 s FWHM) for different averaging periods $\tau$. All three cases converge for averaging periods longer than 100 s.

point in the smoothed dataset would have an uncertainty of $2.1\,\mu\text{mol·mol}^{-1}$ (corresponding to $\tau = 5\,s$), not $0.8\,\mu\text{mol·mol}^{-1}$ (corresponding to $\tau = 1\,s$).

### 4.2 Meteorological conditions

From the data collected during ScaleX 2016 we calculate NEE for the nights 6–7 July and 9–10 July. The sun set at 19:15 UTC and 19:14 UTC on 6 and 9 July, respectively, and rose at 03:25 UTC and 03:27 UTC on 7 and 10 July, respectively. Both nights were free of precipitation. Cloud cover was high during the first night (see Fig. 6a), but the pronounced negative net radiation (Fig. 6b) indicates that the clouds were mostly transparent for outgoing long-wave radiation. In the second night the sky was clearer, resulting in a steadier radiation balance. During both nights, strong radiative cooling was observed. Air temperature decreased from 18 °C to 9 °C and from 24 °C to 11 °C over the course of the first and second night, respectively (Fig. 6c). In combination with low wind speeds (Fig. 6d) this lead to the development of a pronounced temperature inversion at the surface, i.e. a stable NBL. The change from positive to negative net radiation occurs approximately at $t_0 = 18{:}00$ UTC in both nights.

### 4.3 Profiles

We carried out a total of 27 flights during the ScaleX 2016 campaign. For the calculation of a NBL budget we analyse those flights that took place after $t_0 = 18{:}00$ UTC and reached a height of at least 125 m. Twelve flights fulfil these criteria: flight 4 through 10 (first night, Fig. 7) and flight 19 through 23 (second night, Fig. 8). For display in panel b of Fig. 7 and 8 the $CO_2$

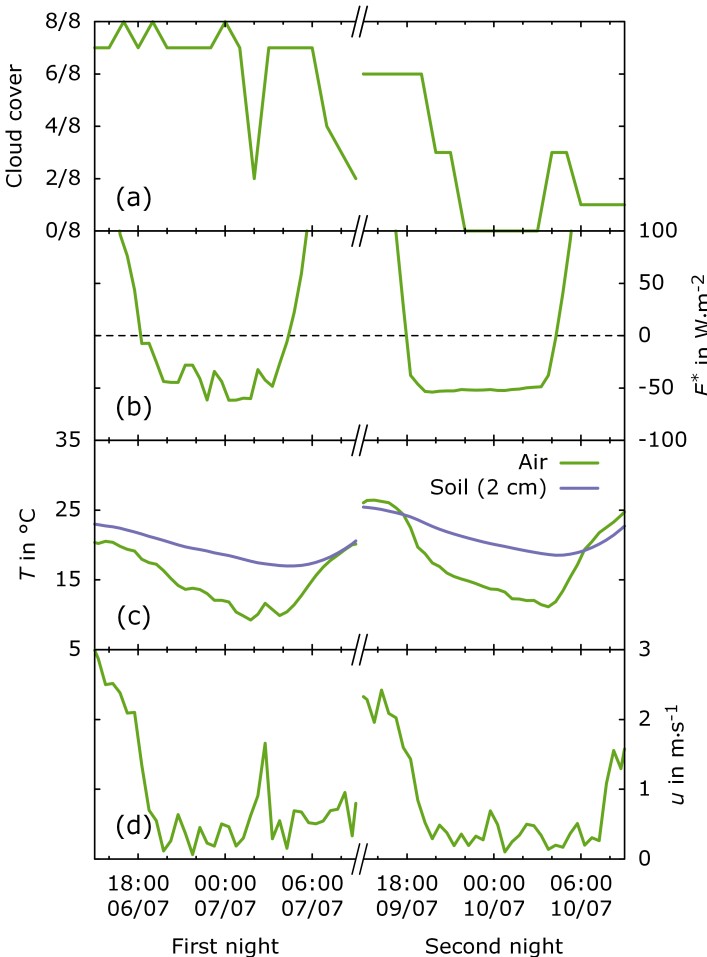

**Figure 6.** Meteorological conditions during the NBL soundings: (a) cloud cover, (b) net radiation $E^*$, (c) temperature $T$ of air (2 m height) and soil (2 cm depth), (d) horizontal wind speed $u$ (3.5 m height). Cloud cover was determined at MOHP, all other observations were made directly at Fendt. Time ist given in UTC.

**Table 2.** Take-off and landing times (UTC)

| Flight | Date | Take-off | Landing | Duration |
|:---:|:---:|:---:|:---:|:---:|
| | year-month-day | hour:minute | hour:minute | |
| #4 | 2016-07-06 | 18:04 | 18:14 | 10 min 50 s |
| #5 | 2016-07-06 | 19:05 | 19:16 | 10 min 58 s |
| #6 | 2016-07-06 | 21:13 | 21:23 | 10 min 13 s |
| #7 | 2016-07-06 | 22:09 | 22:19 | 09 min 20 s |
| #8 | 2016-07-06 | 23:06 | 23:16 | 09 min 43 s |
| #9 | 2016-07-07 | 00:18 | 00:28 | 09 min 23 s |
| #10 | 2016-07-07 | 01:11 | 01:20 | 09 min 18 s |
| #19 | 2016-07-09 | 20:01 | 20:11 | 09 min 58 s |
| #20 | 2016-07-09 | 21:02 | 21:13 | 11 min 31 s |
| #21 | 2016-07-09 | 22:43 | 22:55 | 11 min 23 s |
| #22 | 2016-07-09 | 23:39 | 23:49 | 10 min 34 s |
| #23 | 2016-07-10 | 00:31 | 00:42 | 10 min 32 s |

dry air mole fraction has been smoothed with a Gauss filter of 5 s FWHM. To prevent distortion in the vertical direction, the height above ground level $z$ has been filtered the same way. For this reason, the upper end of the profiles in panel b is at slightly lower height than in panel a. Calculation of the NBL fluxes (see Section 4.5) was carried out with unfiltered $x_{CO2}$ and $z$.

The times given in Fig. 7 and 8 are the midtimes of the flights rounded to full 10 minutes for readability. The exact times of

take-off and landing are provided in Table 2.

During the first night, a stable NBL can be identified from the UAS profiles for flights 6 through 10. The upper end of the temperature inversion aligns with the top of the $CO_2$ enhancement to within 10–20 m. At the time of flights 6 and 8 through 10, the NBL has a height of 50–70 m, whereas the profile from flight 7 indicates a greater NBL height of $\approx 100$ m. We interpret this as an indication that the column measured in flight 7 has been influenced by katabatic inflow of cool, $CO_2$-enriched air at some

point during the night, potentially hours before the flight and kilometers away from Fendt. This interpretation is supported by the flux estimates (see Section 4.5). The profiles from flight 5, which exhibit virtually no gradient, are discussed below.

During the second night, a stable NBL with a height of 50–70 m is visible in all profiles. The flight pattern had been refined and included two ascents and descents far enough from each other to avoid disturbance of the measurements in the second part by air movements caused during the first part. These redundant measurements give insight to the reliability of the measurement

system and to the variability of temperature and $CO_2$ dry air mole fraction on small temporal and spatial scales. The data from flight 21 agrees well between each of the two ascents and descents, suggesting that disturbance by the UAS, instrument noise and drift are small compared to the observed signals. Flight 22 and 23 were carried out only one and two hours later, respectively, and followed the same flight track. However, the data from these flights reveals considerable differences between each of the two ascents and descents, especially in $x_{CO2}$ for heights below 50 m. We interpret this as natural variability on

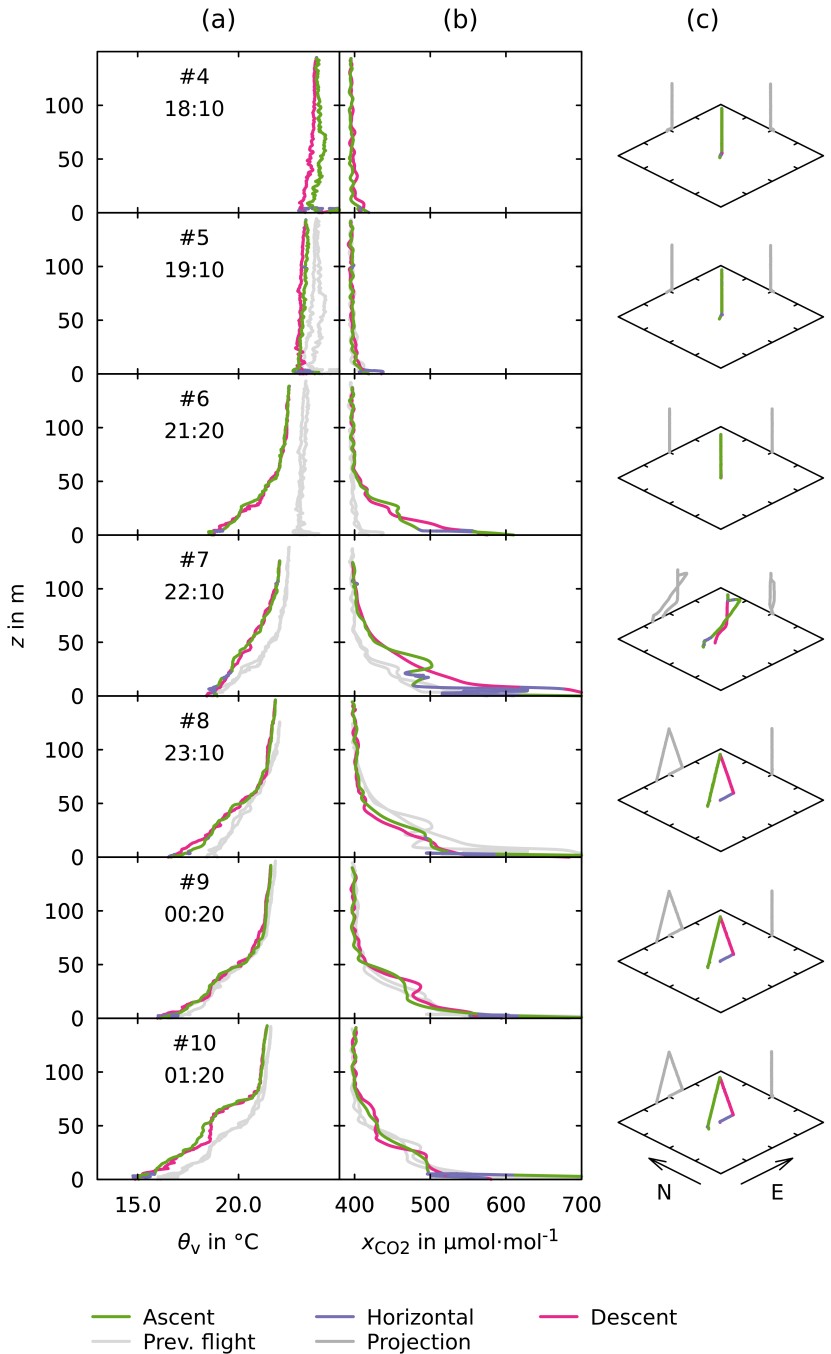

**Figure 7.** Profiles from flights 4 through 10, carried out in the night from 6 to 7 July 2016. Times are given in UTC and specify the middle of the flight, rounded to the next 10 min. (a) Virtual potential temperature $\theta_v$ at height above ground level $z$. (b) Carbon dioxide dry air mole fraction $x_{CO2}$ at height above ground level $z$. To reduce noise $x_{CO2}$ has been smoothed (for details see text). In both (a) and (b) the light grey curves are copies of the previous profile (ascent, horizontal flight and descent combined). (c) Flight track with horizontal projections (grey) for clearness. Northward and eastward direction are marked. The tick marks at the ground plane are 100 m apart. The location of take-off and landing differs slightly between flights, but all flights took place within the same $250{\times}250{\times}150\,\text{m}^3$ bounding box.

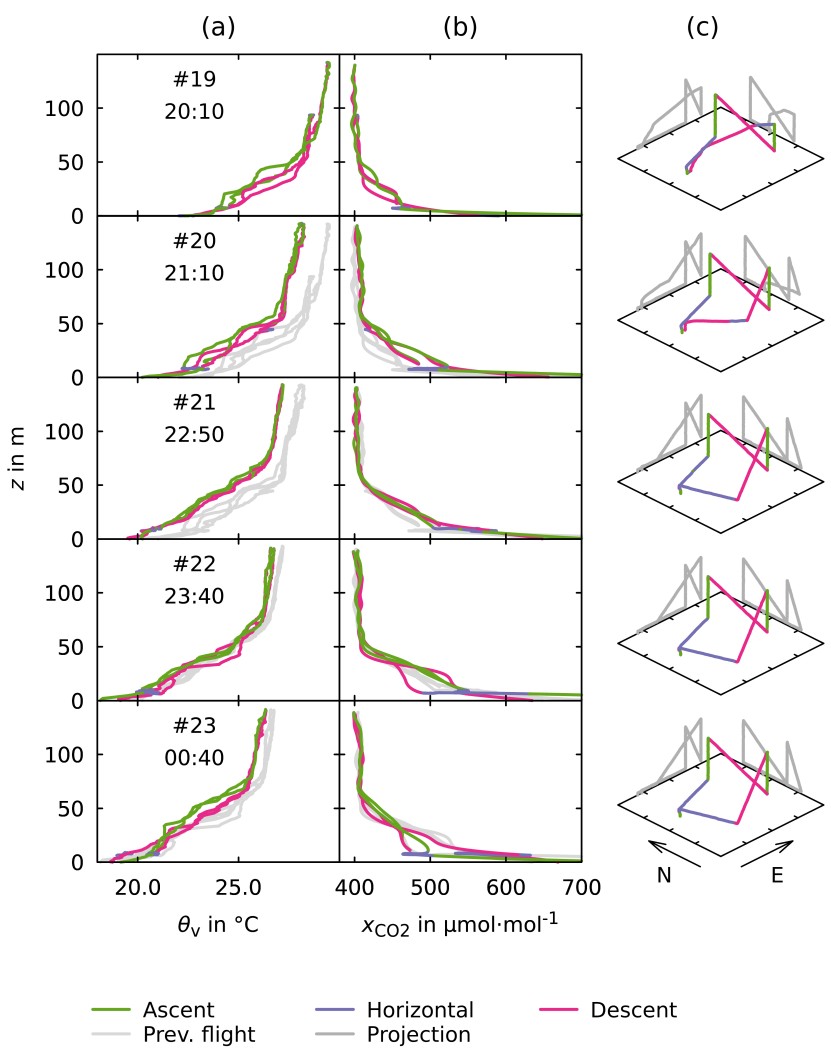

**Figure 8.** Same as Fig. 7, but for flights 19 through 23 carried out in the night from 9 to 10 July 2016. The axis for virtual potential temperature is shifted towards higher temperatures compared to Fig. 7, but covers the same span. All other axes are unchanged. The horizontal legs were flown at 10 m height.

the scale of the flight track, i.e. $\approx 200\,m$ in horizontal distance and $\approx 3\,min$ in time. This small-scale variability is a source of random error in NBL budgets. In our flux calculations multiple ascents during the same flight are effectively averaged, resulting in a reduction of the random error.

The $x_{CO2}$ profiles measured during flight 20, 22 and 23 all exhibit a non-zero gradient with height in the region above the
strong inversion, indicating that some $CO_2$ has escaped the stable NBL. This is supported by the profiles of virtual potential temperature, which are more inclined above the NBL in comparison to the first night. Both features might be the result of intermittent turbulence, a phenomenon often observed at night that can have different causes (see Aubinet (2008) and references therein). Our budgets include the measurements up to 125 m height, so any $CO_2$ that has been transported higher than this is missing in the budgets. In future campaigns, flights with a greater maximum height could be carried out to quantify the effect
this has on the NBL-derived fluxes, or to extend the budget vertically.

The flight pattern used during the second night also included two horizontal transects at 10 m height that were flown at a ground speed of $3\,m\cdot s^{-1}$. Their purpose was to enable measurements of undisturbed air near the ground, but later analysis of flight 14 (see Section 3.1) revealed that the ground speed was insufficient to fully reach this goal.

The profiles for flight 5 are close to straight vertical lines, which would indicate a well-mixed atmosphere. However, they
were measured under low wind speed one full hour after the surface radiation balance became negative, i.e. under conditions favourable for the development of a stable nocturnal boundary layer and accumulation of $CO_2$ near the ground. This apparent contradiction can be explained by comparing COCAP's data to tower-based measurements. Fig. 9a shows the $CO_2$ profile taken by COCAP together with data from the 9 m mast and from HPB (see Sections 2.1 and 2.2). The diagram includes those measurements from the mast that fall into the time interval from 15 min before take-off to 15 min after landing. They reveal
that the $CO_2$ dry air mole fraction near the ground was increased relative to the upper two thirds of the profile and fluctuated strongly, e.g. between 450 and 650 $\mu mol\cdot mol^{-1}$ at 3 m height. These observations are in line with a weakly stable layer near the surface: Surface fluxes accumulated in this layer, but weak turbulent events caused e.g. by wind shear occasionally spread them out to higher layers. The disturbance by the multicopter during take-off or landing prevented COCAP from capturing this accumulation. On the other hand, the higher part of COCAP's profile, taken in the residual layer that is left over from the
daytime mixed layer, matches the mean $CO_2$ dry air mole fraction measured at HPB during the time interval from 1 h before take-off to 1 h after landing. This agreement confirms that COCAP was working properly during the flight.

The profile from flight 8, carried out later in the same night, is consistent with the measurements at the 9 m mast (Fig. 9). We see two reasons for this difference to flight 5. Firstly, the radiative cooling (see Fig. 6) at the time of flight 8 (23:10) was stronger than at the time of flight 5 (19:10). The temperature gradient near the ground was not resolved during flight 5, but
the weaker radiative cooling compared to the later flight has likely resulted in a weaker temperature inversion that allowed more vertical displacement of air by the multicopter. Secondly, the thicker NBL at 23:10 with a less steep $CO_2$ gradient close to the ground means that potential sampling of air parcels originating from above or below the multicopter did not affect the measurements during flight 8 as much as during flight 5.

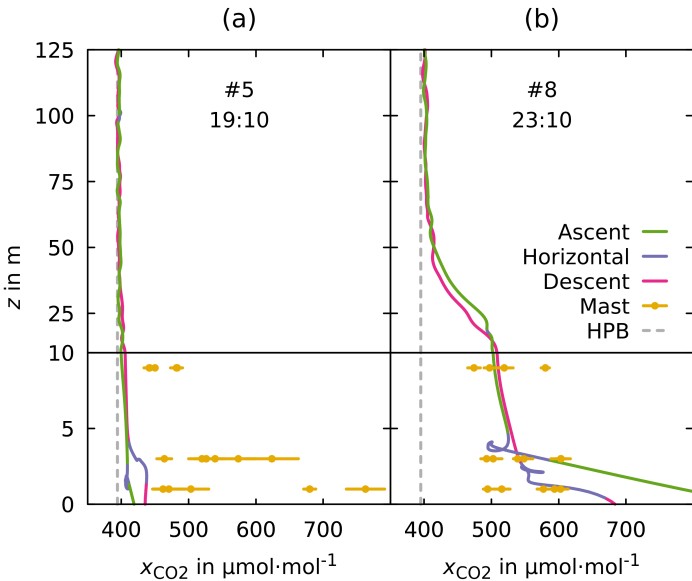

**Figure 9.** Comparison of $x_{CO2}$ measurements by COCAP, by the ICOS station HPB (a point measurement 460 m above ground level at Fendt as described in Section 2.2, representing the $CO_2$ dry air mole fraction in the residual layer) and by the on-site 9 m mast (a) for flight 5 and (b) for flight 8. The dots and bars are mean and standard deviation, respectively, for each 1-minute sampling period of the mast within the time interval from 15 min before take-off to 15 min after landing. Note that the scaling of the vertical axis changes at height $z = 10$ m.

At heights above 70 m the $CO_2$ profile from flight 8 approaches the measurements at HPB, indicating that the stable NBL retains most of the surface emitted $CO_2$. Likewise, the $CO_2$ profiles of all other flights come near the measurements at HPB above the NBL (not shown). This suggests that any transport of $CO_2$ across the top of the NBL is small in magnitude.

The measurement of continuous profiles of the $CO_2$ dry air mole fraction up to heights of 100 m or more has been challenging in the past. In some studies, NBL budgets were therefore based on a measurement near the ground and an assumed gradient up to the top of the NBL. However, the complex shape of the profiles displayed in Fig. 7 and 8 suggest that neither the assumption of a constant (cf. Acevedo et al. (2004)) nor a linearly decreasing (cf. Culf et al. (1997)) $CO_2$ dry air mole fraction would properly represent the conditions at Fendt. The detailed structures resolved in our measurements also indicate great potential of combined measurements of meteorological parameters and trace gas mole fractions for studying small-scale phenomena in the NBL.

**4.4 Disturbance by the UAS**

The potential virtual temperature measured at heights between 10 and 60 m are generally higher during descent than during ascent. This effect is more pronounced for flights 19 through 23 (Fig. 8), likely due to the stronger temperature gradient compared to flights 5 through 10 (Fig. 7). The observed difference supports the reasoning of Section 3.1: As the multicopter

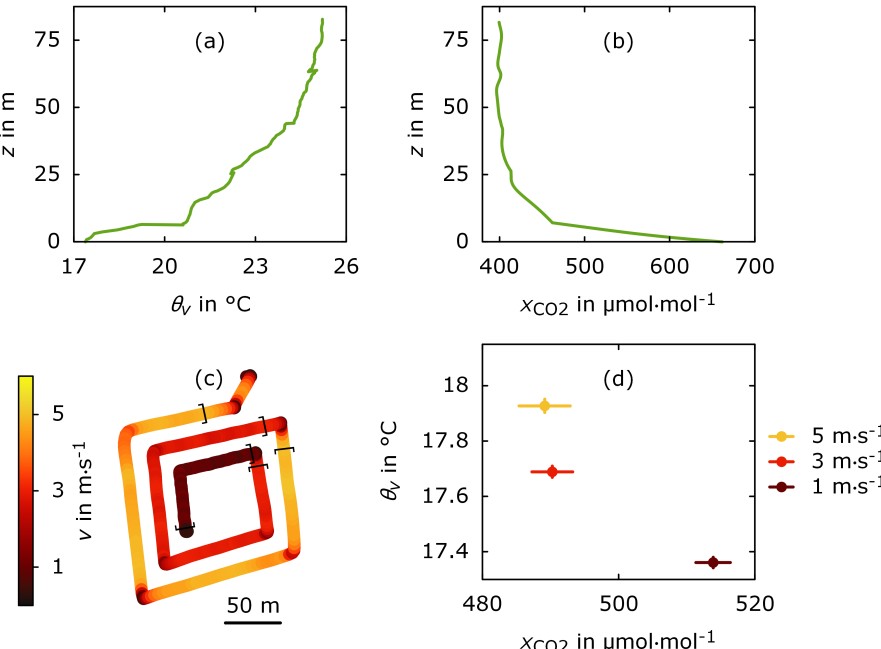

**Figure 10.** Vertical profiles of (a) virtual potential temperature $\theta_v$ and (b) $CO_2$ dry air mole fraction $x_{CO2}$ at 20:15 UTC on 7 July 2016. (c) Track of flight at 22:15 UTC on the same night coloured by horizontal ground speed. Three sections of nominal speed $5\,\text{m·s}^{-1}$, $3\,\text{m·s}^{-1}$ and $1\,\text{m·s}^{-1}$ are marked with brackets. Height was $10\,\text{m}$ above ground throughout the flight. (d) Median of virtual potential temperature and $CO_2$ dry air mole fraction measured during those three sections of the flight. Bars represent the bootstrapped standard error of the median, see text for details. The standard error of the virtual potential temperature is so small that the vertical bars are barely visible. At lower speeds, $\theta_v$ is lower and $x_{CO2}$ higher, suggesting the sampling of air that originates from below the flight height.

descends, the onboard sensors measure warmer air that was pushed downwards by the rotors. Close to the ground (at heights below $10\,\text{m}$) closed flow loops start to form and colder air from below the multicopter reaches the sensors during descent, as can be seen in the profiles from flights 6, 8, 9, 10 and 23.

Systematic differences between ascent and descent are less visible in the profiles of $CO_2$ dry air mole fraction, likely due to
5   a larger variability of $CO_2$ within the nocturnal boundary layer. This variability is reflected in the difference in $x_{CO2}$ between each of the two ascents and descents in the flights 19 through 23, esp. flight 20 and 23.

Flight 14 was dedicated to the investigation of vertical mixing during horizontal movement at different air speeds. It was carried out on 7 July at 22:15 UTC. Winds were particularly low that night (on average $0.3\,\text{m·s}^{-1}$ between 22:00 and 22:30 UTC) and hence ground speed of the UAS was approximately equal to air speed. A stable nocturnal boundary layer had
10  developed, as can be seen from the profiles of $\theta_v$ and $x_{CO2}$ measured during an earlier flight at 20:15 UTC (see Fig. 10 panels a and b). The UAS flew a spiral pattern at a height of $10\,\text{m}$ above ground with decreasing ground speed (Fig. 10c). Throughout flight 14, COCAP's air inlet faced the direction of movement. The flight took place over a flat, homogeneous

meadow. Hence, we assume that terrain and vegetation had caused no heterogeneity in the lateral distribution of temperature and $CO_2$. We analyse three sections of nominal speed $5\,\mathrm{m\cdot s^{-1}}$, $3\,\mathrm{m\cdot s^{-1}}$ and $1\,\mathrm{m\cdot s^{-1}}$. Figure 10d shows the median virtual potential temperature and $CO_2$ dry air mole fraction for each section. The standard error of the median was calculated by bootstrapping with 1000 samples generated from the empirical distribution of the measurements (Wilcox, 2012, pp. 43) and is depicted as horizontal and vertical bars.

The decrease of virtual potential temperature with decreasing speed in Fig. 10d suggests that upward mixing of air from lower layers has a stronger influence on the measurements at lower speed. Likewise, the $CO_2$ dry air mole fraction measured at $1\,\mathrm{m\cdot s^{-1}}$ is $20\,\mathrm{\mu mol\cdot mol^{-1}}$ higher than during faster flight. However, we did not observe a significant difference in $x_{CO2}$ between a ground speed of $3\,\mathrm{m\cdot s^{-1}}$ and $5\,\mathrm{m\cdot s^{-1}}$. The sample inlet for the $CO_2$ measurement extends $20\,\mathrm{cm}$ to the side of the rotors, while temperature and humidity are measured directly below a rotor (see Fig. 3). As the sample inlet was pointing forward throughout the flight, it might have mostly avoided partially closed flow loops during movement at $5\,\mathrm{m\cdot s^{-1}}$, while the temperature and humidity sensors were still affected.

In summary, our results suggest that measurements taken during the ascent of the multicopter are more reliable than those taken during descent and hover. Horizontal transects at low heights can yield measurements that are contaminated with air from below the sampling height. This contamination is lower at higher horizontal air speed, because the multicopter moves away from the vortices it has created. Our experiment does not answer the question whether at $10\,\mathrm{m}$ height a horizontal speed of $5\,\mathrm{m\cdot s^{-1}}$ is sufficient to avoid the contamination entirely.

## 4.5  Carbon dioxide fluxes

The first profiles of the first and second night were taken at 18:10 UTC (flight 4) and 20:10 UTC (flight 19), respectively. Hence, Flight 4 is representative for the $x_{CO2}$ profile at $t_0 = 18:00$ UTC, but flight 19 is not. We therefore need an estimate for the profile at $t_0$. Due to the convective mixing that takes place during the day, the $CO_2$ dry air mole fraction within the boundary layer is nearly independent of height, an assumption that is supported by the profile from flight 4 (see Fig. 7). Assuming further that all surface fluxes were trapped in the developing NBL, air parcels above the NBL height should have preserved the $CO_2$ dry air mole fraction of the column between $t_0$ and the time of the first flight. Consequently, we assume the whole column $x_{CO2}(t_0, z)$ to be equal to the mean dry air mole fraction of the first measured profile between $50\,\mathrm{m}$ and $125\,\mathrm{m}$ height. For consistency we apply this approach to both nights.

The fluxes we calculated from the NBL budgets are listed in Table 3, given as amount of $CO_2$ per time and surface area. The storage flux in Table 3 corresponds to term (A) in Equation 11, the subsidence flux to term (B) and the total flux is equal to $\overline{S}$, i.e. the NEE averaged over the time from $t_0$ to $t_F$. During both nights, horizontal convergence of air masses lead to lifting and consequently a negative subsidence flux. However, the subsidence flux was small compared to the storage flux, accounting for about 1 percent of the total flux. An important consequence of the low subsidence flux is that errors stemming from the simplified model of the NBL growth (see Sect. 3.3) have only a minor influence on the uncertainty of the total flux.

The plausibility of our results can be checked against EC and chamber measurements taken at Fendt. Both the EC and the chamber measurements observed only the fluxes from the pasture at the site, while the NBL budget has a larger footprint. Even

**Table 3.** Fluxes of $CO_2$ calculated from NBL budgets. Begin and end are given in UTC. The end time is specified as the midtime of the portion of the flight used for determination of the NBL budget.

| Begin<br>dd/mm HH:MM | End<br>dd/mm HH:MM (flight) | Storage flux<br>in $\mu mol \cdot m^{-2} \cdot s^{-1}$ | Subsidence flux<br>in $\mu mol \cdot m^{-2} \cdot s^{-1}$ | Total flux<br>in $\mu mol \cdot m^{-2} \cdot s^{-1}$ |
|---|---|---|---|---|
| 06/07 18:00 | 06/07 19:08 (#5) | 13.7 | 0.0 | 13.7 |
| | 06/07 21:16 (#6) | 12.3 | -0.1 | 12.2 |
| | 06/07 22:14 (#7) | 16.1 | -0.2 | 15.9 |
| | 06/07 23:09 (#8) | 11.3 | -0.1 | 11.2 |
| | 07/07 00:21 (#9) | 9.4 | -0.1 | 9.3 |
| | 07/07 01:13 (#10) | 8.4 | -0.1 | 8.4 |
| | | | | |
| 09/07 18:00 | 09/07 21:06 (#20) | 17.2 | -0.2 | 17.0 |
| | 09/07 22:48 (#21) | 11.3 | -0.1 | 11.2 |
| | 09/07 23:43 (#22) | 11 | -0.1 | 10.9 |
| | 10/07 00:35 (#23) | 9.9 | 0.0 | 9.9 |

at low wind speeds of $0.5 \, m \cdot s^{-1}$ air parcels travel 1.8 km every hour. Therefore, the NBL budget also includes sources that are located at several kilometres distance. Given the land cover around Fendt, those sources likely include forests, crop fields and potentially some residential areas (see Fig 14 and 15 for exemplary footprints). Nevertheless, as pasture is the dominant land cover in the area, all three methods should agree in the order of magnitude of the $CO_2$ flux at night.

No EC measurements of acceptable quality are available for either of the nights we probed the NBL (Fig. 11). Conditions of strong radiative cooling combined with weak wind resulted in stable conditions and a violation of the assumptions underlying the EC technique. As a backup, we calculated the mean diurnal cycle from the EC measurements taken between 4 July 2016 00:00 UTC and 11 July 2016 23:59 UTC, a period that includes all our flights and was reasonably consistent in the diurnal variations of temperature. The result is presented in Fig. 11. All fluxes calculated from the NBL budget lie within the range of

NEE observed by EC between 18:00 and 01:00 UTC ($6–16 \, \mu mol \cdot m^{-2} \cdot s^{-1}$). The later the flight at Fendt took place, the lower the NBL-based average NEE, indicating a decreasing flux over the course of the night. We interpret this, at least partially, as an effect of the temperature decrease during the night (Fig. 6), which reduces respiration. In contrast, NEE measured by the EC station increases during the night. However, an increase in respiration over the course of the night is implausible. Given the small number of EC measurements of acceptable quality this apparent trend is likely an artefact.

Figure 12 shows the NBL-derived fluxes in comparison to chamber measurements. Data from the small chambers is available only for the second night. Opaque chambers measure respiration, while clear chambers, the EC station and the NBL budget observe NEE. Therefore, a comparison of the fluxes obtained with these different techniques is only meaningful when photosynthesis is low or absent, i.e. roughly between sunset and sunrise. The convergence of the fluxes of the clear and dark chambers just after 18:00 UTC suggests that photosynthesis has largely ceased as early as $t_0$. Hence, throughout the time span

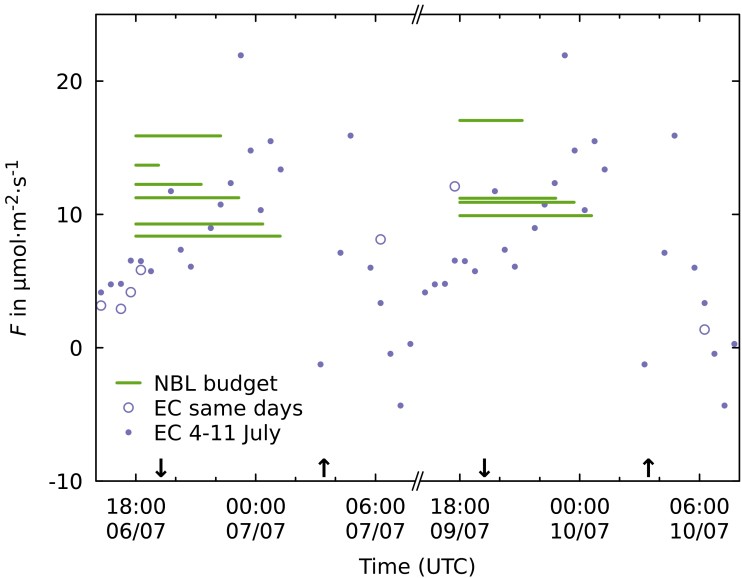

**Figure 11.** Comparison of vertical $CO_2$ fluxes $F$ calculated from the NBL budget and using the EC method. The fluxes from the NBL budget are depicted as lines, where the vertical position of each line represents the average flux over the time span specified by the horizontal extend of the line. Open circles represent the quality-filtered EC measurements taken on the same days as the NBL measurements. Solid dots represent the mean diurnal cycle of the quality-filtered EC measurements averaged over the period from 4 July 2016 00:00 UTC to 11 July 2016 23:59 UTC. Upward and downward arrows mark the time of sunrise and sunset, respectively. NBL and EC agree in magnitude of NEE at night, but not in sign of trend.

for which we determine the NBL budget NEE is dominated by respiration and all the different techniques are comparable. Surprisingly, the measurements with the big chambers yield fluxes only 1/3 as high as obtained with the small chambers, even though all chambers were deployed close to each other on the same meadow. Despite careful investigation the reason for this discrepancy has not yet been found. The NBL budget agrees in magnitude to the fluxes measured with the small chambers.

5 Similarly to the NBL budget, all chamber measurements exhibit a negative trend in fluxes over the course of both nights.

In addition to in situ measurements at Fendt, the range of nighttime NEE of pasture and forests observed in other studies at central European sites with a climate similar to Fendt (Cfb or Dfb in the Köppen-Geiger classification according to Peel et al. (2007)) provides a plausibility check for the NBL budgets (Table 4).

We exclude crop fields from the comparison, as their NEE depends heavily on crop type and time of harvest. Compared

10 to the literature values, NEE for Fendt derived from the NBL budget is on the high end of ranges reported for pasture and higher than most fluxes reported for forests. One explanation is that our measurements took place on two fair weather days in the warmest month of the year 2016, which likely resulted in higher respiration than observed on average over a longer period. Furthermore, Fendt lies in a region with organically rich soils (Fig. 2a). Soil organic carbon content has been shown to be positively correlated with microbial biomass (Habashi, 2016), suggesting particularly strong respiration under beneficial

**Table 4.** Nighttime $CO_2$ fluxes observed in other studies (minimum–maximum of reported values)

| Location | Land cover | Period dd/mm/yyyy | Method | Flux $\mu mol \cdot m^{-2} \cdot s^{-1}$ | Source |
|---|---|---|---|---|---|
| Grillenburg, Germany | Pasture | 02/07/2004–16/07/2004 | EC | 3.9–10.2 | Gilmanov et al. (2007) |
| Stubai Valley, Austria | Pasture | 01/07/2002–30/07/2002 | EC | 6–17 | Wohlfahrt et al. (2005) |
| Stubai Valley, Austria | Pasture | 01/07/2002–30/07/2002 | Chambers + model | 6–11 | Wohlfahrt et al. (2005) |
| Stuttgart, Germany | Pasture | 13/08/2006–17/11/2006 | Chambers | 1.3–3.2 | Chen et al. (2014) |
| Waldstein–Weidenbrunnen, Germany | Managed spruce forest | 01/06/2007–15/07/2007 | Mass balance | 1–7 | Siebicke et al. (2012) |
| Hesse, France | Managed beech forest | 05/08/2005–06/08/2005 | EC | 1.1–5.5 | Longdoz et al. (2008) |
| Hainich, Germany | Unmanaged beech forest | 12/08/2005–30/09/2005 | Mass balance | 3–9 | Kutsch et al. (2008) |
| Hainich, Germany | Unmanaged beech forest | 12/08/2005–30/09/2005 | Bottom-up model | 5–5.5 | Kutsch et al. (2008) |
| Mooseurach, Germany | Managed spruce forest | 01/01/2011–31/12/2011 | EC | -1–15 | Hommeltenberg et al. (2014) |

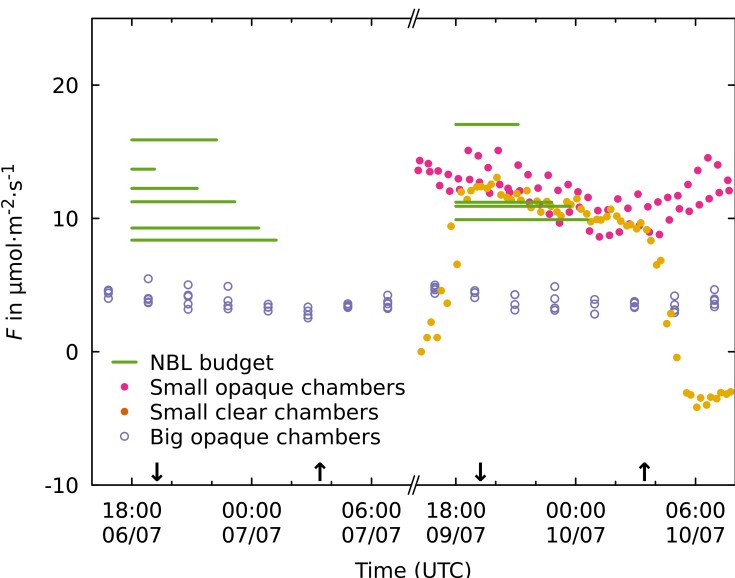

**Figure 12.** Comparison of $CO_2$ flux $F$ calculated from the NBL budget and flux measured with different chambers. The Axis scaling and sunset/sunrise markers are the same as in Fig. 3. Nighttime fluxes observed with the small and the big chambers differ by a factor of three for unknown reasons. The NBL budget agrees in magnitude and sign of trend to the measurements with the small chambers.

conditions. This explanation is supported by the measurements at Mooseurach (Table 4), a drained peatland forest 20 km to the East of Fendt, where respiration fluxes of up to $15\,\mu\text{mol}\cdot\text{m}^{-2}\cdot\text{s}^{-1}$ have been observed.

Another potential cause for higher fluxes observed with the NBL budget relates to the terrain at Fendt. At night, katabic flows of cool, $CO_2$-rich air can stream down the steep slope west of the measurement site. Though Fendt is situated in a valley with only a shallow slope to the east, this inflow might lead to localised lifting of air that is not accounted for in the ECMWF IFS data and hence not included in our calculation of subsidence. The increased NBL height and high variability in the lowest 50 m observed during flight 7 as well as the higher flux derived from the NBL budget are an indication of such an inflow event.

### 4.6 Sensitivity of fluxes

The NBL budget is influenced by measurement uncertainty, incomplete knowledge about the state of the atmosphere and data selection. In order to quantitatively assess the influence of these factors on our results, we changed the procedure of calculating fluxes in either of the following ways:

1. by adding a bias of $\pm 2$ m to the altitude measurements,

2. by adding a bias of $\pm 3\,\mu\text{mol}\cdot\text{mol}^{-1}$ to $x_{CO2}$ of all but the first profile of each night,

**Table 5.** Sensitivity of $CO_2$ flux to different factors (see text for details). Unit of fluxes is $\mu mol \cdot m^{-2} \cdot s^{-1}$.

| Night | No change | $z + 2\,m$ | $z - 2\,m$ | $x_{CO2}+$ 3 ppm | $x_{CO2}-$ 3 ppm | Whole flight | w/o 9 m mast | No subsid. |
|---|---|---|---|---|---|---|---|---|
| 1 | 11.8 | 12.2 | 10.6 | 13.1 | 10.5 | 11.4 | 10.8 | 11.9 |
| | | (+3 %) | (-10 %) | (+11 %) | (-11 %) | (-3 %) | (-8 %) | (+1 %) |
| 2 | 12.3 | 12.7 | 11.7 | 13.1 | 11.4 | 11.9 | 12.7 | 12.4 |
| | | (+3 %) | (-5 %) | (+7 %) | (-7 %) | (-3 %) | (+3 %) | (+1 %) |

3. by using COCAP data for the whole column instead of replacing $x_{CO2}$ in the lowest 9 m with measurements taken at the 9 m mast,

4. by using COCAP data taken during the whole flight, i.e. using ascent, descent and hover instead of ascent only, or

5. by disregarding subsidence.

Check 1 accounts for the uncertainty of COCAP's pressure-based altitude measurements. Check 2 allows us to evaluate the influence of both the uncertainty of COCAP's $x_{CO2}$ measurements and the spatial heterogeneity of $x(t_0)$. The former is known from experiment (see Section 4.1) and the latter can be estimated from the $CO_2$ measurements at HPB. Assuming that the 131 m inlet at HPB is in the residual layer all night, the interquartile range of the $x_{CO2}$ measurements of a single night approximately reflects the variability of the background onto which fluxes accumulate. It amounts to $1.1\,\mu mol \cdot mol^{-1}$ and $2.4\,\mu mol \cdot mol^{-1}$ for the period from 18:00 UTC to 02:00 UTC in the first and second night, respectively. Check 3 and 4 relate to the disturbance caused by the UAS, which is discussed in Sections 3.1 and 4.4.

The mean fluxes for each night obtained using the changed procedures are summarised in Table 5. The largest difference to the normal ("No change") procedure occurs when $x_{CO2}$ is altered ($\pm 11\%$ for the first and $\pm 7\%$ for the second night, respectively). Changing the altitude or not using the data from the 9 m mast also have a considerable influence on the mean flux.

Fig. 13 shows the values from Table 5 in graphical form. In addition, the fluxes calculated for each flight are depicted, visualising how their spread is affected by the different checks. A substantial increase in spread is observed only when the data from the 9 m mast is not used.

Overall, the results from the sensitivity checks indicate that the NBL method is robust against measurement uncertainty in the altitude and $x_{CO2}$ measurements, spatial heterogeneity of $x(t_0)$, disturbance of the NBL caused by the UAS and the effect of subsidence. It should be noted that the mean vertical wind extracted from the ECMWF IFS model was relatively small during the two nights of our measurements. Under different conditions, e.g. in a strong high pressure system, the effect of subsidence or lifting on the NBL budget could be much higher.

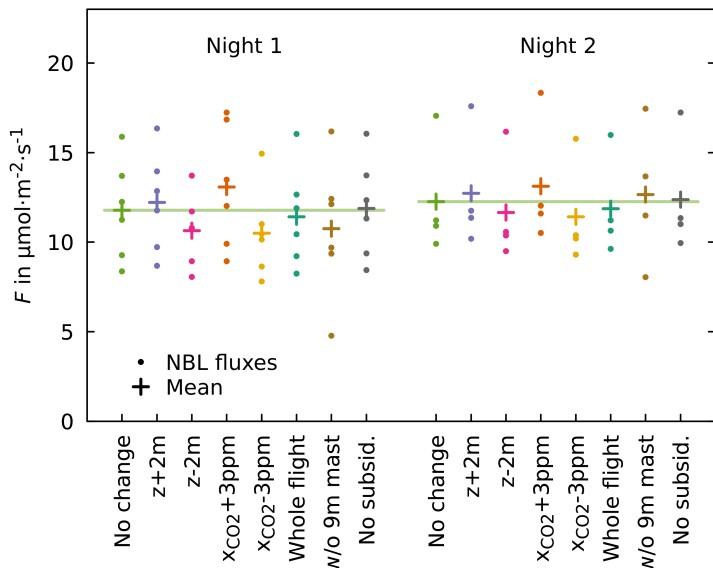

**Figure 13.** Sensitivity of $CO_2$ fluxes calculated from the NBL budgets to changes in calculation procedure. Dots denote the fluxes calculated for each flight, crosses mark the mean of these fluxes. The lines are visual aids to facilitate comparison to the mean flux of the normal ("No change") procedure.

## 4.7 Flux footprint

Example flux footprints of one NBL budget of each night are visualised in Fig. 14 and Fig. 15.

The footprint depicted in Fig. 14 was calculated for a column of air passing Fendt on 6 July 2016 at 21:00 UTC, i.e. close to the time of flight 6. The 1% contour of the footprint encloses an area of $60\,km^2$, which accounts for 60 % of the total sensitivity.

The land cover map suggests that the NBL budget represents mainly the respiration of forests, pasture and crop lands north of Fendt, with little contribution from urban areas.

The footprint depicted in Fig. 15 was calculated for a column of air passing Fendt on 9 July 2016 at 21:00 UTC, i.e. close to the time of flight 20. The 1% contour of the footprint encloses an area of $80\,km^2$, which accounts for 70 % of the total sensitivity. Again, the NBL budget is mainly influenced by forests, pasture and crop lands.

The footprints for other times during the two nights are similar in size, i.e. on the order of $100\,km^2$. They mostly cover the sector within 20 km north-west to north-east of Fendt.

We recognise that the relatively low spatial and temporal resolution of the ECMWF IFS meteorological model entails errors in the transport modelling. Variability of the horizontal wind component within a grid cell and on time scales below three hours is neglected, possibly resulting in an underestimation of the footprint size. Likewise, terrain features that are smaller than a

grid cell are not represented in the meteorological model. However, as our NBL budgets cover time scales of 1–7 hours and the

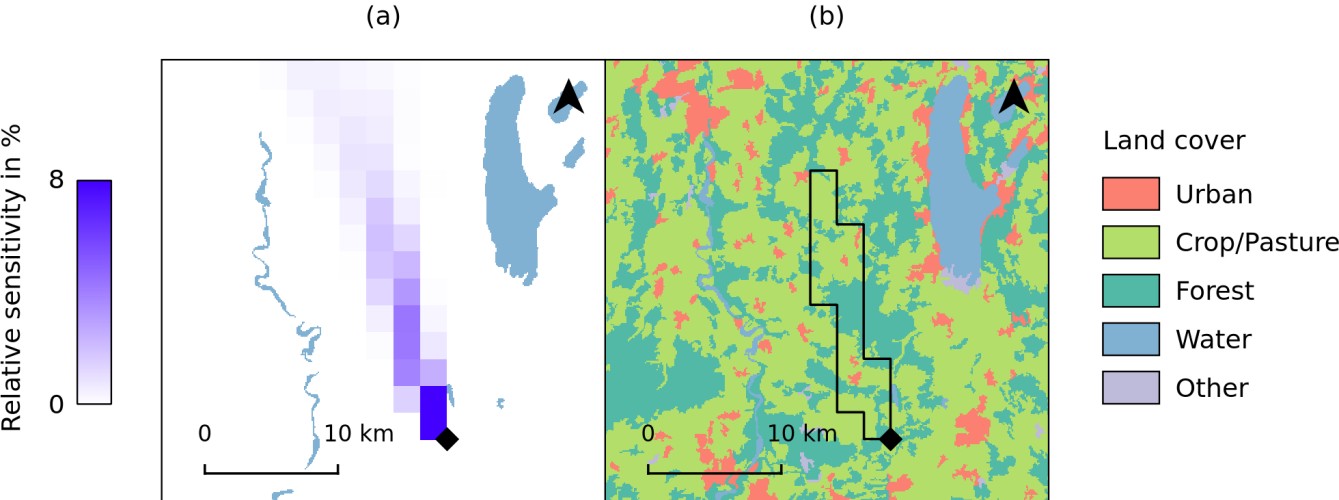

**Figure 14.** Footprint of an NBL budget at Fendt on 6 July 21:00 UTC. (a) Relative contribution of each grid cell to the total sensitivity of the budget to surface fluxes. Water bodies depicted for orientation. (b) Contour of all grid cells with a relative sensitivity of 1% or higher on top of a simplified land cover map (CORINE 2012 v18.5, European Environment Agency, EEA (2016)). The area observed is dominated by forests, pasture and crop land.

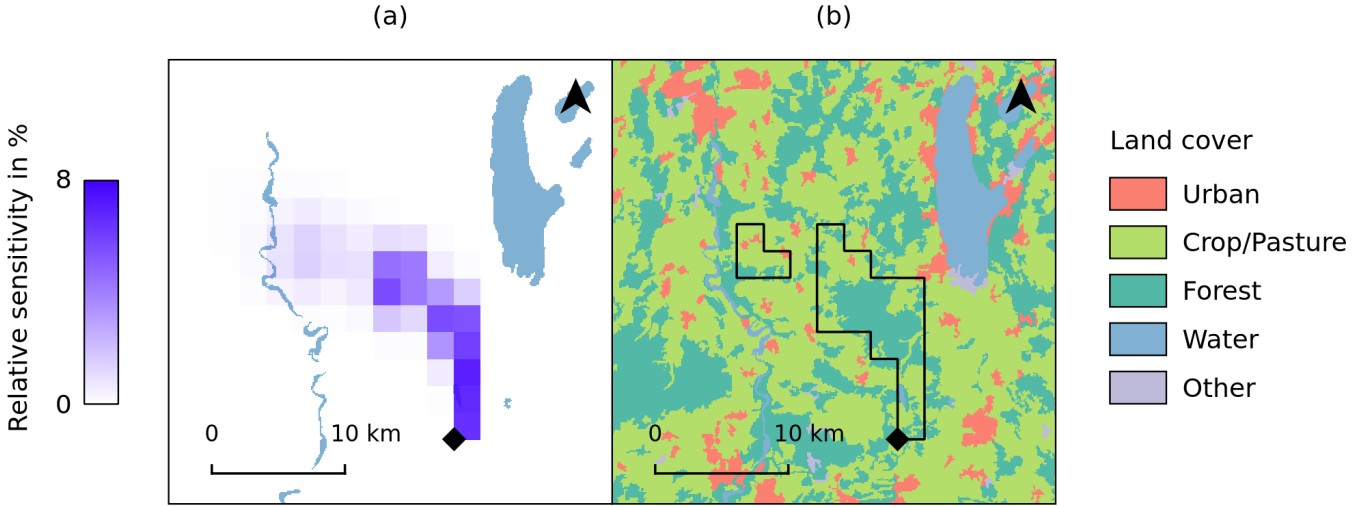

**Figure 15.** Same as Fig. 14, but for 9 July 21:00 UTC

footprints extend over many grid cells, sub-scale variability should play only a minor role. We are therefore confident that that the model results provide a reasonable estimate of the region seen by the NBL budget method.

## 5  Conclusions and outlook

To the best of our knowledge, we have for the first time determined nocturnal boundary layer budgets based on trace gas measurements with an unmanned aircraft. During two nights we repeatedly sampled the NBL with a multicopter carrying COCAP, a lightweight analyser designed for deployment on unmanned aircraft. Simultaneous measurement of $CO_2$ dry air mole fraction, air temperature, humidity and pressure allowed to quantify the rate of accumulation of carbon dioxide in the NBL. By applying deconvolution we could improve the temporal resolution of the $CO_2$ measurements, thus achieving a higher vertical resolution of the profiles. We estimated the effect of subsidence or lifting on the NBL budgets with the help of weather forecast data and corrected the budgets accordingly. The respiration fluxes obtained from the NBL budgets are plausible in comparison to other flux measurements at the Fendt site, though on the high end of the range reported in the literature for sites with land cover and climatic conditions similar to Fendt. A potential positive bias in the obtained fluxes could be caused by convergence of cool, $CO_2$-rich air at the floor of the valley in which Fendt is located. The current data set does not allow to confirm or rule out this effect. In a future campaign, however, simultaneous deployment of a second UAS on the elevated plateau west of the site could provide more insight, as downward transport of $CO_2$ should result in consistently lower accumulation and hence lower flux estimates obtained from NBL budgets on the plateau.

We have investigated how the disturbance of the NBL caused by a multicopter influences in situ measurements. We found that while flying close to the ground, air from below the UAS can reach the sensors, causing a bias if the respective quantity has a non-zero gradient. To prevent this bias from affecting the NBL budget we replaced the airborne $x_{CO2}$ measurements taken at low height with measurements from a 9 m mast. At greater height, some of our profiles exhibit a systematic difference between ascent and descent. During descent, the airborne sensors are moved into a volume of air that may have been disturbed by the downwash of the multicopter's rotors. Therefore, we use only data captured during ascent for NBL budgeting.

The robustness of our approach has been demonstrated by a sensitivity analysis. The largest uncertainty of the NBL budget is caused by spatial heterogeneity of the $CO_2$ dry air mole fraction in the late afternoon combined with the uncertainty of the $CO_2$ measurement. The estimated combined error in $x_{CO2}$ results in $\pm 11\%$ change of the mean of the fluxes obtained from the NBL budget for the first night. Using only data from the UAS and not from the 9 m mast increased the spread of the fluxes, but changed their mean by no more than $10\%$. This suggests that satisfactory NBL budgets can be determined from UAS data even if no stationary measurements near the ground are available. For future studies, we suggest to position the sample inlet 50–100 cm above the rotors to further reduce the sampling of air that was displaced or mixed by the UAS.

The region that influences an NBL budget has often not been reported in past studies. We improved on this situation by carrying out mesoscale modelling. While the driving meteorological data and the underlying topography do not resolve small structures at and below the scale of 1 km, our method gives at least an estimate of the region that influences the NBL budget. Under the conditions of our measurements the footprints were on the order of $100\,\mathrm{km}^2$ in size. In situ wind measurements

would enable validation of the meteorological data and possibly improvement of the transport modelling. Such measurements could be taken by UAS without the need for additional sensors (Mayer et al., 2012; Neumann and Bartholmai, 2015).

Future NBL studies could employ multiple UASs simultaneously to quantify spatial heterogeneity and horizontal gradients in the $CO_2$ dry air mole fraction. Firstly, this would support the analysis of the uncertainty of the NBL-derived fluxes. Secondly, concurrent profiles could yield constraints for the net advection of $CO_2$.

While we carried out our measurements with multicopters, fixed-wing aircraft would also be capable platforms for NBL studies. The vortices generated by their wings are slower and spread out wider than the concentrated downwash produced by the rotors of a multicopter. Therefore they should cause less interference with the NBL soundings and could provide precise measurements down to ground level. Additionally, their typically higher horizontal speed makes it easier to evade any disturbance that they create.

Another possibility to reduce the disturbance of measurements near the ground would be a different placement of the inlet. Given the asymmetric flow pattern below and above a multicopter's rotors (see Section 3.1), sampling from several rotor diameters above the UAS should reduce the artefacts caused by closed flow loops.

NBL budgets based on UAS measurements are an effective and efficient tool for the quantification of nocturnal fluxes. Besides ecosystem respiration, it could also be applied to detect carbon dioxide emissions of other sources, e.g. urban areas. Small and lightweight sensors for other tracers such as methane would open up even more possibilities. Alternatively, compact time-resolved sampling systems (Andersen et al., 2017) or long flexible tubing (Brosy et al., 2017) can be used in connection with conventional ground-based instrumentation to measure a whole range of species.

In summary, we have demonstrated that nocturnal surface flux estimates can be derived from UAS-based gas measurements by means of an NBL budget approach. Given the moderate cost of UAS and their minimal infrastructure requirements this innovation makes the NBL budget method for the quantification of surface fluxes much more accessible. Spurred by the increasing adoption of unmanned aircraft in geoscience and the development of miniaturized high-accuracy sensors for different tracers we foresee wide adoption of this technique in the coming years.

*Code and data availability.* Measurement data from the UAS, output of the STILT model, analysis scripts and instructions how to run them are available at https://dx.doi.org/10.17617/3.3g . Time series of cloudiness observed at MOHB can be downloaded from the Climate Data Center (https://opendata.dwd.de/climate_environment/CDC/). Time series of the $CO_2$ dry air mole fraction measured at HPB can be requested from DWD; contact: dagmar.kubistin@dwd.de

*Author contributions.* R.H.G., M.K. and J.L. conceptualised and carried out the UAS-based measurements and M.K. curated the data obtained. M.Z. coordinated the ScaleX campaign, operated the EC station and the small chamber measurements and curated the data obtained. R.G. operated the big chamber measurements and curated the data obtained. B.W. operated the $x_{CO2}$ measurements at the 9 m mast and curated the data obtained. M.S. was responsible for the operation of the ICOS HPB site and curated the data obtained. C.G. and F.-T. K. ran

the STILT model. C.G., R.H.G., M.K. and J.L. analysed the data. M.K. wrote the original draft of this publication. All authors reviewed the draft. M.K. compiled the final manuscript.

*Competing interests.*  The authors declare that they have no conflict of interest.

*Acknowledgements.*  We thank the scientific teams of the ScaleX Campaign 2016 for their contribution, especially Klaus Schäfer for coordination of work package 2, Caroline Brosy for taking care of the flight permissions and John E. Flatt for piloting the UAS. Thanks to the staff of DWD for providing meteorological and trace gas data from Hohenpeißenberg. We thank Ingo Völksch for soil data of the Fendt site. M.K. thanks Fanny Kittler for illuminative discussions about the EC method. We gratefully acknowledge the authors of various open-source software packages that were used in our study and for the preparation of the manuscript, in particular Inkscape, GIMP, GNU Octave (Eaton et al., 2017), gnuplot, LaTeX, LyX and QGIS. The colours in diagrams and maps are based on the work of Cynthia A. Brewer (Brewer, 2017), Peter Kovesi (Kovesi, 2015) and the Wikicarto 2.0 colour map (Wikipedia contributors, 2012). We thank the Max Planck Society for generous financial support. The TERrestrial Environmental Observatory (TERENO) pre-Alpine research is funded by the Helmholtz Association ATMO Programme and the Federal Ministry of Education and Research. M.Z. received support from the German Research Foundation (DFG; grant ZE 1006/2-1).

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
