# Peer review of "Surface flux estimates derived from UAS-based mole fraction measurements by means of a nocturnal boundary layer budget approach"

_Atmospheric Measurement Techniques, 2019_

## Referee Comment (RC1) · Anonymous Referee #2 · 24 Sep 2019

General comments:

Overall, this is an excellent and exciting paper. It demonstrates a novel application of UAS for atmospheric science and adds to an exciting literature concerning the new horizons this sampling platform offers. It is a proof-of-concept study, intended to demonstrate the potential use of UAS in CO2 biospheric respiration measurement. It identifies the challenge and importance of nocturnal respiration measurements and the gap that EC methods (and limited spatial scale of chambers) cannot fill. It proposes a mass balancing approach suited to night-time measurement, taking advantage of the assumption of a stable boundary layer. Given that this is an initial study, intended to open up

a new direction in this field, some of the questions about the validity of the flux method itself (see specific comments) should be seen in that context, i.e. that this paper identifies a problem and suggests an innovative approach that can be built on and refined in future work. I believe the paper would be of great interest to readers of AMT and the quality of presentation, figures etc is excellent. I specifically praise the way the authors have carefully considered the specific challenges of rotary UAS sampling (i.e. the influence of downwash, instrument response time, etc) and proposes a solution to only use descent profiles to avoid disturbance and take into account response time. These factors are often overlooked and this paper serves as excellent guidance. The paper also compares UAS results with chambers and raises some interesting questions.

I do have some important comments though. These concern the UAS flux approach and the way in which surface footprint and vertical mixing scales have been derived (see specific comments). I hope that these comments can be addressed or answered in a revised version of the paper. I see this method as something that can be improved upon in future work and perhaps the most important edits to the text could highlight the remaining uncertainties and challenges to the approach.

Specific Comments:

1/ Use of STILT to define footprint: I have sympathy with the approach and I do not have a good alternative solution to accurate night-time footprint evaluation, however Lagrangian trajectories near to the surface are known to be subject to significant error/uncertainty. Surface trajectories tend to hug the surface and follow (typically) the 10 m wind vector suggested in the reanalysis met data used to drive the model (in this case ECMWF 0.1 IFS), i.e. upward/downward motions are supressed. How many vertical levels does this version of ECMWF have and what resolution in the vertical domain used in the study? The approach used here is to release 10000 particles per time-step at very small increments in height up to some assumed mixing height (see comment below). I would raise some concerns with this approach. Perhaps an improvement may be to run STILT in ensemble mode – to perturb each trajectory with

some assigned uncertainty (diagnosed from the ECMWF data or obtained by drone-based wind measurement variability in future) to the wind vector to examine advective uncertainty - Section 5 nicely acknowledges the future role of wind measurement. A set of releases at different heights is unlikely to recreate any meaningful 2D footprint as the trajectories will cluster along one singular wind vector (as Figure 13 tends to show) extracted from the ECMWF model grid (0.1 deg is $\sim$ 10km of fetch after all), whereas an ensemble may at least give a better qualitative indication of the possible extremes of the fetch/footprint. This is likely to be the biggest source of uncertainty in any Lagrangian budgeting approach and I think it may be important to state this in the paper, even if an ensemble approach is not used in any revision. I realise that footprinting is extremely difficult but it would be useful to acknowledge just how difficult and error-prone it is. The same is true of EC footprints in topographically-variable environments of course.

2/ P.12 line 1 – why is it expected that "Surface fluxes are expected to be diluted into a column that extends from the surface to 1/2 this height in each time step"? This seems rather arbitrary? Why is this expected? How was it derived from ECMWF data? In a stable night-time boundary layer, what is the vertical mixing process assumed to reach this quantitative mixing-height value? In stable NBLs, vertical dilution is dominated by diffusion with some small residual vertical turbulence, e.g. the "fanning" Pasquill stability class. Given that assumed vertical mixing timescales (and horizontal foot-print) are key to deriving flux per unit area in the footprint using the proposed method, these quantities are key. This (and comments below and above) cause me to start to question the overall flux method as it stands. Wouldn't a much more conceptual and simple approach simply be to look at the temporal gradient in CO2 throughout the NBL throughout the night and assume a fetch equivalent to the length scale of advection over that timescale (e.g. treating the NBL like a large-scale vented flux chamber, so long as footprint can be defined)? Such a concept would negate a diagnosis of any spatial heterogeneity in flux (arriving at a bulk net flux for a defined airmass volume) but I don't have any confidence that the proposed approach can do anything better

than this in reality (without a fleet of drones that is). In summary, I'm not convinced that any useful 2D footprint can be obtained, so averaging the accumulated NBL mass over any surface area is problematic, so a simplified NBL bulk net flux approach may be more meaningful?

3/ Other sources of flux uncertainty: These include the assumed background CO2, any variability in upwind sources of CO2 (i.e. variability in the background airmass entering the footprint over the time frame of the measurements), measurement error/precision, wind speed and direction variability etc. Section 4.2 and 4.6 addresses measurement error nicely and explores sensitivity, but not the other sources of flux error. Perhaps it would be good to note these in the paper, even if they cannot be determined or budgeted in this work, so that others following or improving on the work are aware.

4/ Use of w from ECMWF and the nature of night-time lifting or subsidence (page 11): I'm not sure that large scale vertical motions need to be considered in the proposed flux approach. The effect of subsidence is to suppress the night-time boundary layer, i.e. to move the night-time inversion lower. Lifting would act to lift the inversion and entrain air from above (diluting the NBL and therefore XCO2). Since this approach treats the NBL as a flux chamber (in effect), this motion seems not to be important and implicit (i.e. manifest) in the concentration measurements themselves. Or have I interpreted this incorrectly?

Technical comments:

Remember to add spaces between quantities and units (e.g. 100km^2 on line 11) and other instances.

---

## Referee Comment (RC2) · Anonymous Referee #1 · 26 Sep 2019

General: The ms is focused on the application of multi rotor drones and custom build $CO_2$ sensors to estimate nocturnal fluxes and storage in the lower boundary layer. This is a new application of a promising tool and a potential solution to a stability issue in flux measurements that is problematic to EC measurements and the budgetary numbers that we can provide during night-time. Nice work ! I have a few issues that in my opinion could strengthen the ms at this stage; As the authors also conclude, the flux estimates using the NBL seem high and more background information on the site could be useful to assess if the estimates are too high. Information like soil type and organic content as well as NEE flux during the day- time could help in this context, as well as the storage term calculated from the 9 m profile tower at the site. Since this is a well

know methodology, but used in a new context it is of cause important to add credibility from as many other sources as possible, especially since the chamber measurements are quite ambiguous. The instrumental setup seem to work well and fine, but I miss arguments for choosing a custom-made gas analyzer over those relatively cheap and light commercially available analyzers in the market, like e.g. LiCor Li-840 or others.

Specific:

P2 L5: I would assume that sporadic turbulent events would be measured by EC but not molecular diffusion, please consider rephrasing.

P2 L7: you could mention storage estimates by use of concentration profiles in a tower, could be mentioned.

P3 L:31: please provide crop type and vegetation stage.

P7 L13: It could give the impression that a tower of a considerable height is needed in addition to the UAV approach, is that so? Please specify

P11 L22: I guess if you could assume that day and night time fluxes were even in magnitude, you wouldn't have to measure the night. Consider rephrasing – order of magnitude maybe?

P22 L27: it is well known that chamber measurements can give quite different fluxes within short distances, and since the small are only available part of the time it would make sense to try to establish the storage term of the tower, for comparison.

Fig. 12 I'm not sure this increases the confidence in the method because it basically show a very wide range of possible flux during the two nights.

P27 L4 check fig numbers

---

## Author Comment (AC2) · 18 Dec 2019

**Response to Anonymous Referee #1**

We thank the referee for their thoughtful comments on our manuscript.

Below, all comments are repeated in italics, followed by our response typeset upright. Changes to the manuscript are highlighted in blue colour.

**General comments**

*The ms is focused on the application of multi rotor drones and custom build CO2 sensors to estimate nocturnal fluxes and storage in the lower boundary layer. This is a new application of a promising tool and a potential solution to a stability issue in flux measurements that is problematic to EC measurements and the budgetary numbers that we can provide during night-time. Nice work ! I have a few issues that in my opinion could strengthen the ms at this stage; As the authors also conclude, the flux estimates using the NBL seem high and more background information on the site could be useful to assess if the estimates are too high. Information like soil type and organic content as well as NEE flux during the day- time could help in this context, as well as the storage term calculated from the 9 m profile tower at the site. Since this is a well know methodology, but used in a new context it is of cause important to add credibility from as many other sources as possible, especially since the chamber measurements are quite ambiguous.*

We added a soil type and land cover map (see Fig. 1) as well as the following description to Sect. 2.1 of the manuscript: 'While soil identification at the Fendt site resulted in Stagnosols at three locations, soil organic carbon (SOC) content was determined additionally at 20 locations within a regular grid. SOC content in 5cm depth varied between 4 and 11% at 5 cm depth, while at 50cm depth, values of up to 23% were obtained. The highest SOC contents were observed at the eastern side of the regular grid where a peat area is located. According to BGR (2013), organically rich soils (Cambisols and Histosols) prevail within 20 km radius around the Fendt site (Fig. 2a). The dominant land cover in this region are crops, pasture and forest (Fig. 2b).'

Additionally, we added measurement results from Mooseurach, a drained peatland forest site just 20 km to the East of Fendt (Hommeltenberg et al., 2014) to Table 3 and the following discussion to Sect. 4.5: 'Furthermore, Fendt lies in a region with organically rich soils (Fig. 2a). Soil organic carbon content has been shown to be positively correlated with microbial biomass (Habashi, 2016), suggesting particularly strong respiration under beneficial conditions. This explanation is supported by the measurements at Mooseurach (Table 3), a drained peatland forest 20 km to the East of Fendt, where respiration fluxes of up to $15\,\mu\text{mol}\cdot\text{m}^{-2}\cdot\text{s}^{-1}$ have been observed.'

NEE at Fendt measured by the EC station during July 2016 (Fig. 2) can exceed $10\,\mu\text{mol}\cdot\text{m}^{-2}\cdot\text{s}^{-1}$ at night and $-20\,\mu\text{mol}\cdot\text{m}^{-2}\cdot\text{s}^{-1}$ during the day. The mean nocturnal NEE is close to $8\,\mu\text{mol}\cdot\text{m}^{-2}\cdot\text{s}^{-1}$, but this is an average over different cut-and-collect management stages and weather scenarios during July. The high temperature and high soil moisture conditions at the time of our NBL measurements are not well represented in this average.

IMK-IFU runs another EC station at a grassland site near Rottenbuch, located approximately 12 km south-west of the Fendt site. NEE at the Rottenbuch site is on the same order of magnitude as the fluxes observed at the Fendt site (Zeeman et al., 2017).

The storage term calculated from the 9 m mast is already part of our NBL-budgets (as described in Sect. 3.1). Furthermore, Fig. 6 and 7 show that the accumulation of $CO_2$ takes place up to a height of 50–80 m, i.e. the 9 m mast can measure only an unknown fraction of the total storage. For these two reasons we think that storage fluxes calculated from the 9 m mast cannot serve as reference for the NBL-derived fluxes.

Taking all available evidence together, the NBL-derived flux estimates do not seem too high.

*The instrumental setup seem to work well and fine, but I miss arguments for choosing a custom-made gas analyzer over those relatively cheap and light commercially available analyzers in the market, like e.g. LiCor Li-840 or others.*

The specific requirements for a $CO_2$ analyser for unmanned aircraft and how COCAP meets them is detailed in Kunz et al. (2018), cited in Sect. 2.3 where we explain our setup. Interested readers can therefore easily get this background information and we would rather not reiterate it in this manuscript. In comparison to the LI-840 it should be noted that both instruments weigh around 1 kg, but COCAP contains sensors for ambient temperature, pressure and humidity, a data logger, a pump, a flow controller, as well as a radio for realtime data transmission, all of which are missing in the LI-840. Moreover, the effects of rapid changes in temperature and pressure (as they occur during UAS flights, but not in laboratory deployment) on the LI-840's $x_{CO_2}$ measurements would need to be evaluated before using it in this application.

**Specific comments**

We notice that the reviewer refers to version 1 of the manuscript, which was updated based on suggestions by the handling editor before the discussion phase started. Hence, the line numbers are slightly offset with respect to the discussion paper.

*P2 L5: I would assume that sporadic turbulent events would be measured by EC but not molecular diffusion, please consider rephrasing.*

Our intent here was to describe the roots of the EC nighttime problem in one sentence, but this likely resulted in oversimplification. Instead of substantially increasing the length of this paragraph, we now refer the reader to a text book: 'Stable conditions violate assumptions underlying the EC technique (see Aubinet et al., 2012 for a comprehensive discussion).'

Molecular diffusion is negligible for atmospheric transport on the scale of meters (Lee et al., 2005) and therefore not mentioned here.

*P2 L7: you could mention storage estimates by use of concentration profiles in a tower, could be mentioned.*

See above for small structures like the 9 m mast in Fendt. Utilizing a tall tower for obtaining nighttime NEE estimates is mentioned on p. 2 ll. 31–33.

*P3 L:31: please provide crop type and vegetation stage.*

We extended the first paragraph of Sect. 2.1: 'The valley floor is dominated by pasture and some crops, predominantly maize, which in Germany is typically sowed in April or May and harvested between September and November.'

*P7 L13: It could give the impression that a tower of a considerable height is needed in addition to the UAV approach, is that so? Please specify*

In our study we made use of the $CO_2$ dry air mole fraction measurements of an instrumented 9 m mast. However, in Sect. 4.6 we present fluxes calculated by using COCAP data only, i.e. disregarding the measurements at the mast. The spread of the fluxes increases, but the mean flux changes by only 8 % for the first and 3 % for the second night, hence a mast is not strictly necessary. We made the respective lines in Sect. 3.1 clearer: 'Furthermore we discard COCAP's $x_{CO_2}$ data collected below 9 m height for the calculation of the NBL budget. Instead, the lowest part of the $x_{CO_2}$ profile is defined by the stationary measurements at the 9 m mast at 1, 3 and 9 m height. Pressure and temperature at these levels are interpolated from COCAP's measurements. During flight, the horizontal distance between COCAP and the 9 m mast was lower than 150 m at any time. Hence, we do not expect pronounced horizontal gradients in $x_{CO_2}$ between the measurement locations. In Sect. 4.6 we discuss how the NBL-derived fluxes are affected if the data from the 9 m mast is not used.'

*P11 L22: I guess if you could assume that day and night time fluxes were even in magnitude, you wouldn't have to measure the night. Consider rephrasing – order of magnitude maybe?*

Thank you for this suggestion. We changed the respective sentence to 'The sign of the daytime $CO_2$ flux is generally negative, whereas the sign of the nighttime flux is positive, but they are usually of the same order of magnitude.'.

*P22 L27: it is well known that chamber measurements can give quite different fluxes within short distances, and since the small are only available part of the time it would make sense to try to establish the storage term of the tower, for comparison.*

We agree that spatial heterogeneity can lead to large differences in enclosure-based flux measurements. It is unclear, however, why agreement is high among all small chambers as well as among all big chambers, but poor between them. For the reasons explained above the storage term from the 9 m mast cannot serve as an independent reference.

*Fig. 12 I'm not sure this increases the confidence in the method because it basically show a very wide range of possible flux during the two nights.*

Each green horizontal line in Fig. 10 and 11 corresponds to one of the green dots in Fig. 12 ('No change'), so their spread is exactly the same. Part of the flux variability is a negative trend during the night, which might be a real phenomenon caused by temperature, as explained on p. 22, ll. 16–18. Moreover, the footprint of the NBL budgets changes over time, meaning that different areas with higher or lower respiration contribute to the NBL budgets with changing proportion, leading to another physical cause for variability in the NBL-derived fluxes.

In the sensitivity analysis the only substantial increase in spread occurs when the measurements from the 9 m mast are not used. Even in that case the small change in mean flux indicates that little or no bias is introduced. We see Fig. 12 as a valuable and honest depiction of the uncertainty of the NBL-derived fluxes. The repositioning of the air inlet suggested in the Conclusions might well be able to reduce the spread in the flux estimates.

*P27 L4 check fig numbers*

Corrected (already in the discussion paper).

**References**

Aubinet, M., Feigenwinter, C., Heinesch, B., Laffineur, Q., Papale, D., Reichstein, M., Rinne, J., and Van Gorsel, E.: Nighttime Flux Correction, in: Eddy Covariance: A Practical Guide to Measurement and Data Analysis, edited by Aubinet, M., Vesala, T., and Papale, D., Springer Atmospheric Sciences, pp. 133–157, Springer Netherlands, Dordrecht, doi:10.1007/978-94-007-2351-1_5, 2012.

BGR: Bodenübersichtskarte Der Bundesrepublik Deutschland 1:1.000.000 (BÜK1000DE), URL https://produktcenter.bgr.de/terraCatalog/DetailResult.do?fileIdentifier= A95A723E-1274-4601-9E60-27079436F1F3, 2013.

EEA: CLC 2012 — Copernicus Land Monitoring Service, URL http://land.copernicus.eu/ pan-european/corine-land-cover/clc-2012, 2016.

Habashi, H.: Effect of Forest and Soil Type on Microbial Biomass Carbon and Respiration, Eurasian Soil Science, 49, 1084–1089, doi:10.1134/S1064229316090064, 2016.

Hommeltenberg, J., Schmid, H. P., Drösler, M., and Werle, P.: Can a Bog Drained for Forestry Be a Stronger Carbon Sink than a Natural Bog Forest?, Biogeosciences, 11, 3477–3493, doi:https://doi.org/10.5194/bg-11-3477-2014, 2014.

IUSS Working Group WRB: World Reference Base for Soil Resources 2014, Update 2015. International Soil Classification System for Naming Soils and Creating Legends for Soil Maps, no. 106 in World Soil Resources Reports, FAO, Rome, 2015.

Lee, X., Finnigan, J., and Paw U, K. T.: Coordinate Systems and Flux Bias Error, in: Handbook of Micrometeorology: A Guide for Surface Flux Measurement and Analysis, edited by Lee, X., Massman, W., and Law, B., Atmospheric and Oceanographic Sciences Library, pp. 33–66, Springer Netherlands, Dordrecht, doi:10.1007/1-4020-2265-4_3, 2005.

Zeeman, M. J., Mauder, M., Steinbrecher, R., Heidbach, K., Eckart, E., and Schmid, H. P.: Reduced Snow Cover Affects Productivity of Upland Temperate Grasslands, Agricultural and Forest Meteorology, 232, 514–526, doi:10.1016/j.agrformet.2016.09.002, 2017.

[Figure]

Figure 1: (a) Soil types in the region around the Fendt site, based on BGR (2013), denoted in WRB classification (IUSS Working Group WRB, 2015). 'Fl./Gl.' stands for 'Fluvisols/Gleysols'. (b) Simplified land cover map (CORINE 2012 v18.5, European Environment Agency, EEA (2016)) of the same region. In both panels the location of the Fendt site is marked with a black diamond.

[Figure]

Figure 2: Statistics of NEE fluxes obtained with the EC technique during July 2016

---

## Author Response (AR1)

**Author comments to review of amt-2019-221**

15 December 2019

We thank the two referees for their thoughtful comments on the manuscript. Below, all comments are repeated in italics, followed by our response typeset upright. Changes to the manuscript are highlighted in blue colour. The list of references and new figures are included below our responses.

A marked-up version of the revised manuscript is attached.

**Comments from Anonymous Referee #1**

**General comments**

*The ms is focused on the application of multi rotor drones and custom build CO2 sensors to estimate nocturnal fluxes and storage in the lower boundary layer. This is a new application of a promising tool and a potential solution to a stability issue in flux measurements that is problematic to EC measurements and the budgetary numbers that we can provide during night-time. Nice work ! I have a few issues that in my opinion could strengthen the ms at this stage; As the authors also conclude, the flux estimates using the NBL seem high and more background information on the site could be useful to assess if the estimates are too high. Information like soil type and organic content as well as NEE flux during the day- time could help in this context, as well as the storage term calculated from the 9 m profile tower at the site. Since this is a well know methodology, but used in a new context it is of cause important to add credibility from as many other sources as possible, especially since the chamber measurements are quite ambiguous.*

We added a soil type and land cover map (see Fig. 1) as well as the following description to Sect. 2.1 of the manuscript: 'While soil identification at the Fendt site resulted in Stagnosols at three locations, soil organic carbon (SOC) content was determined additionally at 20 locations within a regular grid. SOC content in 5cm depth varied between 4 and 11% at 5 cm depth, while at 50cm depth, values of up to 23% were obtained. The highest SOC contents were observed at the eastern side of the regular grid where a peat area is located. According to BGR (2013), organically rich soils (Cambisols and Histosols) prevail within 20 km radius around the Fendt site (Fig. 2a). The dominant land cover in this region are crops, pasture and forest (Fig. 2b).'

Additionally, we added measurement results from Mooseurach, a drained peatland forest site just 20 km to the East of Fendt (Hommeltenberg et al., 2014) to Table 3 and the following discussion to Sect. 4.5: 'Furthermore, Fendt lies in a region with organically rich soils (Fig. 2a). Soil organic carbon content has been shown to be positively correlated with microbial biomass (Habashi, 2016), suggesting particularly strong respiration under beneficial conditions. This explanation is supported by the measurements at Mooseurach (Table 3), a drained peatland forest 20 km to the East of Fendt, where respiration fluxes of up to $15\,\mu mol{\cdot}m^{-2}{\cdot}s^{-1}$ have been observed.'

NEE at Fendt measured by the EC station during July 2016 (Fig. 2) can exceed $10\,\mathrm{\mu mol} \cdot$ $\mathrm{m}^{-2} \cdot \mathrm{s}^{-1}$ at night and -20 $\mathrm{\mu mol} \cdot \mathrm{m}^{-2} \cdot \mathrm{s}^{-1}$ during the day. The mean nocturnal NEE is close to $8\,\mathrm{\mu mol} \cdot \mathrm{m}^{-2} \cdot \mathrm{s}^{-1}$, but this is an average over different cut-and-collect management stages and weather scenarios during July. The high temperature and high soil moisture conditions at the time of our NBL measurements are not well represented in this average.

IMK-IFU runs another EC station at a grassland site near Rottenbuch, located approximately 12 km south-west of the Fendt site. NEE at the Rottenbuch site is on the same order of magnitude as the fluxes observed at the Fendt site (Zeeman et al., 2017).

The storage term calculated from the 9 m mast is already part of our NBL-budgets (as described in Sect. 3.1). Furthermore, Fig. 6 and 7 show that the accumulation of $CO_2$ takes place up to a height of 50–80 m, i.e. the 9 m mast can measure only an unknown fraction of the total storage. For these two reasons we think that storage fluxes calculated from the 9 m mast cannot serve as reference for the NBL-derived fluxes.

Taking all available evidence together, the NBL-derived flux estimates do not seem too high.

*The instrumental setup seem to work well and fine, but I miss arguments for choosing a custom-made gas analyzer over those relatively cheap and light commercially available analyzers in the market, like e.g. LiCor Li-840 or others.*

The specific requirements for a $CO_2$ analyser for unmanned aircraft and how COCAP meets them is detailed in Kunz et al. (2018), cited in Sect. 2.3 where we explain our setup. Interested readers can therefore easily get this background information and we would rather not reiterate it in this manuscript. In comparison to the LI-840 it should be noted that both instruments weigh around 1 kg, but COCAP contains sensors for ambient temperature, pressure and humidity, a data logger, a pump, a flow controller, as well as a radio for realtime data transmission, all of which are missing in the LI-840. Moreover, the effects of rapid changes in temperature and pressure (as they occur during UAS flights, but not in laboratory deployment) on the LI-840's $x_{CO_2}$ measurements would need to be evaluated before using it in this application.

**Specific comments**

We notice that the reviewer refers to version 1 of the manuscript, which was updated based on suggestions by the handling editor before the discussion phase started. Hence, the line numbers are slightly offset with respect to the discussion paper.

*P2 L5: I would assume that sporadic turbulent events would be measured by EC but not molecular diffusion, please consider rephrasing.*

Our intent here was to describe the roots of the EC nighttime problem in one sentence, but this likely resulted in oversimplification. Instead of substantially increasing the length of this paragraph, we now refer the reader to a text book: 'Stable conditions violate assumptions underlying the EC technique (see Aubinet et al., 2012 for a comprehensive discussion).'

Molecular diffusion is negligible for atmospheric transport on the scale of meters (Lee et al., 2005) and therefore not mentioned here.

*P2 L7: you could mention storage estimates by use of concentration profiles in a tower, could be mentioned.*

See above for small structures like the 9 m mast in Fendt. Utilizing a tall tower for obtaining nighttime NEE estimates is mentioned on p. 2 ll. 31–33.

*P3 L:31: please provide crop type and vegetation stage.*

We extended the first paragraph of Sect. 2.1: 'The valley floor is dominated by pasture and some crops, predominantly maize, which in Germany is typically sowed in April or May and harvested between September and November.'

*P7 L13: It could give the impression that a tower of a considerable height is needed in addition to the UAV approach, is that so? Please specify*

In our study we made use of the $CO_2$ dry air mole fraction measurements of an instrumented 9 m mast. However, in Sect. 4.6 we present fluxes calculated by using COCAP data only, i.e. disregarding the measurements at the mast. The spread of the fluxes increases, but the mean flux changes by only 8 % for the first and 3 % for the second night, hence a mast is not strictly necessary. We made the respective lines in Sect. 3.1 clearer: 'Furthermore we discard COCAP's $x_{CO_2}$ data collected below 9 m height for the calculation of the NBL budget. Instead, the lowest part of the $x_{CO_2}$ profile is defined by the stationary measurements at the 9 m mast at 1, 3 and 9 m height. Pressure and temperature at these levels are interpolated from COCAP's measurements. During flight, the horizontal distance between COCAP and the 9 m mast was lower than 150 m at any time. Hence, we do not expect pronounced horizontal gradients in $x_{CO_2}$ between the measurement locations. In Sect. 4.6 we discuss how the NBL-derived fluxes are affected if the data from the 9 m mast is not used.'

*P11 L22: I guess if you could assume that day and night time fluxes were even in magnitude, you wouldn't have to measure the night. Consider rephrasing – order of magnitude maybe?*

Thank you for this suggestion. We changed the respective sentence to 'The sign of the daytime $CO_2$ flux is generally negative, whereas the sign of the nighttime flux is positive, but they are usually of the same order of magnitude.'.

*P22 L27: it is well known that chamber measurements can give quite different fluxes within short distances, and since the small are only available part of the time it would make sense to try to establish the storage term of the tower, for comparison.*

We agree that spatial heterogeneity can lead to large differences in enclosure-based flux measurements. It is unclear, however, why agreement is high among all small chambers as well as among all big chambers, but poor between them. For the reasons explained above the storage term from the 9 m mast cannot serve as an independent reference.

*Fig. 12 I'm not sure this increases the confidence in the method because it basically show a very wide range of possible flux during the two nights.*

Each green horizontal line in Fig. 10 and 11 corresponds to one of the green dots in Fig. 12 ('No change'), so their spread is exactly the same. Part of the flux variability is a negative trend during the night, which might be a real phenomenon caused by temperature, as explained on p. 22, ll. 16–18. Moreover, the footprint of the NBL budgets changes over time, meaning that different areas with higher or lower respiration contribute to the NBL budgets with changing proportion, leading to another physical cause for variability in the NBL-derived fluxes.

In the sensitivity analysis the only substantial increase in spread occurs when the measurements from the 9 m mast are not used. Even in that case the small change in mean flux indicates that little or no bias is introduced. We see Fig. 12 as a valuable and honest depiction of the uncertainty of the NBL-derived fluxes. The repositioning of the air inlet suggested in the Conclusions might well be able to reduce the spread in the flux estimates.

*P27 L4 check fig numbers*

Corrected (already in the discussion paper).

**Comments from Anonymous Referee #2**

**General comments**

*Overall, this is an excellent and exciting paper. It demonstrates a novel application of UAS for atmospheric science and adds to an exciting literature concerning the new horizons this sampling*

*platform offers. It is a proof-of-concept study, intended to demonstrate the potential use of UAS in CO2 biospheric respiration measurement. It identifies the challenge and importance of nocturnal respiration measurements and the gap that EC methods (and limited spatial scale of chambers) cannot fill. It proposes a mass balancing approach suited to night-time measurement, taking advantage of the assumption of a stable boundary layer. Given that this is an initial study, intended to open up a new direction in this field, some of the questions about the validity of the flux method itself (see specific comments) should be seen in that context, i.e. that this paper identifies a problem and suggests an innovative approach that can be built on and refined in future work. I believe the paper would be of great interest to readers of AMT and the quality of presentation, figures etc is excellent. I specifically praise the way the authors have carefully considered the specific challenges of rotary UAS sampling (i.e. the influence of downwash, instrument response time, etc) and proposes a solution to only use descent profiles to avoid disturbance and take into account response time. These factors are often overlooked and this paper serves as excellent guidance. The paper also compares UAS results with chambers and raises some interesting questions. I do have some important comments though. These concern the UAS flux approach and the way in which surface footprint and vertical mixing scales have been derived (see specific comments). I hope that these comments can be addressed or answered in a revised version of the paper. I see this method as something that can be improved upon in future work and perhaps the most important edits to the text could highlight the remaining uncertainties and challenges to the approach.*

We appreciate that the referee sees our work as a valuable contribution to the scientific community. We share the view that the pilot study presented in our manuscript cannot fully answer all questions about the accuracy of the derived fluxes and associated footprints, but is a foundation that future works can build upon.

**Specific Comments**

*1/ Use of STILT to define footprint: I have sympathy with the approach and I do not have a good alternative solution to accurate night-time footprint evaluation, however Lagrangian trajectories near to the surface are known to be subject to significant error/uncertainty. Surface trajectories tend to hug the surface and follow (typically) the 10 m wind vector suggested in the reanalysis met data used to drive the model (in this case ECMWF 0.1 IFS), i.e. upward/downward motions are supressed. How many vertical levels does this version of ECMWF have and what resolution in the vertical domain used in the study?*

The vertical resolution of the meteorological data is specified in the original manuscript at the end of Sect. 3.4: 'The vertical resolution of the meteorological data depends on height above ground. The lowest layer extends from the ground to 10 m height, the following 5 layers extend from the top of the next lower layer to 31 m, 55 m, 80 m, 108 m and 138 m, respectively.' The total number of layers is 89. In the revised manuscript we add the likewise relevant fact that 'The temporal resolution of the ECMWF IFS data is 3 h.'.

In order to test the hypothesis that vertical motions are suppressed in the transport model, we plotted the height of 200 out of the 10 000 particles as they travel backwards from the Fendt site. As an example we chose the particles released inside the nocturnal boundary layer at $z = 10$ m and particles released above the nocturnal boundary layer at $z = 100$ m on July 6 21:00 UTC (Fig. 3 and 4) and July 9 21:00 UTC (Fig. 5 and 6). Note that these are a small subset of the particle trajectories based on which the footprints presented in Fig. 13 and 14 in the discussion paper were calculated.

A prominent feature of the trajectories for 6 July is the absence of a stable boundary layer for travel distances greater than 5 km, which approximately corresponds to the time period before

20:00 UTC. In contrast, the trajectories for 9 July indicate a stable boundary layer throughout the time period between 18:00 UTC and 21:00 UTC. This would mean that accumulation in a shallow layer near the ground would have happened during a period of 1 h on 6 July and during a period of 3 h on 9 July. While the profiles measured by COCAP (Fig. 6 and 7 in the discussion paper) do suggest a weaker inversion layer on 6 July, a threefold difference in the accumulation seems too high. Additionally, the measurements at the 9 m mast presented in Fig. 8 in the discussion paper show that $CO_2$ has accumulated near the ground as early as 19:10 UTC on 6 July, an observation that cannot be explained without a stable boundary layer. This demonstrates that meteorological datasets are an imperfect description of the atmosphere and more generally underlines the referee's point that transport modelling in the nocturnal boundary layer is subject to considerable uncertainties.

The hypothesis that STILT suppresses vertical motion, however, is clearly refuted by Fig. 3 through 6. Even within the strong inversion on 9 July the particles are frequently redistributed between the two lowermost layers of the meteorological dataset.

*The approach used here is to release 10000 particles per time-step at very small increments in height up to some assumed mixing height (see comment below). I would raise some concerns with this approach. Perhaps an improvement may be to run STILT in ensemble mode – to perturb each trajectory with some assigned uncertainty (diagnosed from the ECMWF data or obtained by drone-based wind measurement variability in future) to the wind vector to examine advective uncertainty - Section 5 nicely acknowledges the future role of wind measurement. A set of releases at different heights is unlikely to recreate any meaningful 2D footprint as the trajectories will cluster along one singular wind vector (as Figure 13 tends to show) extracted from the ECMWF model grid (0.1 deg is ∼ 10km of fetch after all), whereas an ensemble may at least give a better qualitative indication of the possible extremes of the fetch/footprint. This is likely to be the biggest source of uncertainty in any Lagrangian budgeting approach and I think it may be important to state this in the paper, even if an ensemble approach is not used in any revision.*

The STILT model is stochastic in the sense that different realisations of vertical turbulent transport are used for each of the particles released, which is illustrated by Fig. 3 through 6. The particles reside at different vertical levels of the meteorological dataset for different periods of time, hence experiencing different advective transport. The horizontal dispersion resulting from this mechanism can be seen in Fig. 7. Over a travel distance of 10 km these 10 randomly chosen trajectories for 6 July and 9 July spread out over 1.5 km and 5 km, respectively. We do not have a reference at hand to compare to, but this spreading seems reasonable for a statically stable boundary layer. STILT makes use of the atmospheric stability as well as the wind variability between grid cells, vertical levels and time steps. We do not know which further uncertainty could be extracted from the meteorological dataset alone. On the other hand we fully agree that wind measurements alongside the NBL profiling would provide an estimate of the model error and could be used to determine the uncertainty of the footprints.

*I realise that footprinting is extremely difficult but it would be useful to acknowledge just how difficult and error- prone it is. The same is true of EC footprints in topographically-variable environments of course.*

We agree that the calculation of footprints is challenging, especially at the scale relevant for the NBL budgets. The discussion paper lists the limitations of our approach in Sect. 3.4, 4.7 and 5. We add to Sect. 4.7: 'Variability of the horizontal wind component within a grid cell and on time scales below three hours is neglected, possibly resulting in an underestimation of the footprint size. Likewise, terrain features that are smaller than a grid cell are not represented in the meteorological model.'

*2/ P.12 line 1 – why is it expected that "Surface fluxes are expected to be diluted into a column*

*that extends from the surface to 1/2 this height in each time step"? This seems rather arbitrary? Why is this expected? How was it derived from ECMWF data?*

The respective paragraph describes how the STILT model represents vertical mixing. Like any model, it uses a simplified description of reality. In the case depicted in Fig. 5, for example, the STILT model has determined a boundary layer height of approximately 30 m. Whenever a particle resides below 1/2 this height, i.e. below 15 m, it is considered to be in contact with surface fluxes and a finite sensitivity to surface fluxes is assigned to this point of the particle's trajectory. The boundary layer height is calculated using a modified Richardson number method (Lin et al., 2003). The threshold 1/2 has been chosen for computational efficiency; thresholds between 10 % and 100 % of the boundary layer height have been found to have insignificant effects on the footprints (Gerbig et al., 2003). We revised the description in Sect. 3.4: 'To do so, the height up to which mixing occurs is estimated from the meteorological data using a modified Richardson number method (Lin et al., 2003). Surface fluxes influence air parcels within a column that extends from the surface to 1/2 this height in each time step (Gerbig et al., 2003).'

*In a stable night-time boundary layer, what is the vertical mixing process assumed to reach this quantitative mixing-height value? In stable NBLs, vertical dilution is dominated by diffusion with some small residual vertical turbulence, e.g. the "fanning" Pasquill stability class.*

The profiles depicted in Fig. 6 and 7 illustrate that vertical mixing does take place within a stable boundary layer. Profile #20, for example, taken on 9 July at 21:10 UTC, exhibits pronounced gradients both in virtual potential temperature and $CO_2$ dry air mole fraction up to a height of $z = 50\,\mathrm{m}$. Both rapid radiative cooling and the emission of $CO_2$ take place at the ground, so these gradients are the signature of vertical mixing. Molecular diffusion is slow; a $CO_2$ molecule in air at a temperature of 20 °C travels on average 1.6 m in a whole day (Karion et al., 2010). The main mechanism for vertical mixing in the stable NBL is turbulence generated by vertical wind shear due to friction at the surface.

*Given that assumed vertical mixing timescales (and horizontal footprint) are key to deriving flux per unit area in the footprint using the proposed method, these quantities are key. This (and comments below and above) cause me to start to question the overall flux method as it stands.*

This is a misunderstanding. The fluxes are derived from the NBL budgets by means of Equation 11 in the discussion paper, which does not contain any quantity that depends on the calculated footprint. The flux footprint is calculated from Equation 14 and is unitless. The value in each grid cell of the footprint is a measure of how much the flux in this grid cell influenced the flux derived from the NBL budget at the Fendt site.

*Wouldn't a much more conceptual and simple approach simply be to look at the temporal gradient in CO2 throughout the NBL throughout the night and assume a fetch equivalent to the length scale of advection over that timescale (e.g. treating the NBL like a large-scale vented flux chamber, so long as footprint can be defined)? Such a concept would negate a diagnosis of any spatial heterogeneity in flux (arriving at a bulk net flux for a defined airmass volume) but I don't have any confidence that the proposed approach can do anything better than this in reality (without a fleet of drones that is). In summary, I'm not convinced that any useful 2D footprint can be obtained, so averaging the accumulated NBL mass over any surface area is problematic, so a simplified NBL bulk net flux approach may be more meaningful?*

The fluxes derived from NBL budgets are spatial and temporal averages, but this averaging takes place due to the transport in the atmosphere and the physical accumulation in the NBL. We do not apply averaging in the data processing. The role of the footprints is flux attribution to an area, not flux calculation. We agree that transport modelling is subject to errors and it is especially challenging in case of a shallow stable boundary layer. However, as the ECMWF IFS data used to drive the STILT model contains horizontally, vertically and temporally resolved

wind vectors, we are confident that our approach yields a more realistic footprint than an estimate based on a single mean wind vector.

*3/ Other sources of flux uncertainty: These include the assumed background CO2, any variability in upwind sources of CO2 (i.e. variability in the background airmass entering the footprint over the time frame of the measurements), measurement error/precision, wind speed and direction variability etc. Section 4.2 and 4.6 addresses measurement error nicely and explores sensitivity, but not the other sources of flux error. Perhaps it would be good to note these in the paper, even if they cannot be determined or budgeted in this work, so that others following or improving on the work are aware.*

The two sources for uncertainty of the assumed background $CO_2$ dry air mole fraction are measurement error (covered by sensitivity check 1) and spatial variability. The flux error stemming from spatial variability of the background air mass is covered by sensitivity check 2 as detailed in Sect. 4.6. Subgrid and sub-time-step variability of the wind are not represented in the model, except for the vertical turbulence parametrisation. We state this more clearly in Sect. 4.7 of the revised manuscript as detailed above.

*4/ Use of w from ECMWF and the nature of night-time lifting or subsidence (page 11): I'm not sure that large scale vertical motions need to be considered in the proposed flux approach. The effect of subsidence is to suppress the night-time boundary layer, i.e. to move the night-time inversion lower. Lifting would act to lift the inversion and entrain air from above (diluting the NBL and therefore XCO2). Since this approach treats the NBL as a flux chamber (in effect), this motion seems not to be important and implicit (i.e. manifest) in the concentration measurements themselves. Or have I interpreted this incorrectly?*

Subsidence (lifting) is intrinsically tied to horizontal divergence (convergence) of air, which does affect the NBL budgets. Imagine a case with flat terrain and no advection at the measurement location. Without subsidence or lifting, the NBL-derived flux equals the surface flux at the measurement site. If lifting takes place, surface emissions originating from the vicinity of the site 'pile up' at the measurement location and the flux estimates will be too high. Taking into account that subsidence and lifting are relatively slow processes, we do not expect strong mixing and entrainment at the border between the NBL and the residual layer above. Conversely, in case of subsidence, some fraction of the local emissions are dispersed horizontally and not included in the NBL budget, resulting in too low flux estimates.

**Technical comments**

*Remember to add spaces between quantities and units (e.g. 100km^2 on line 11) and other instances.*

We use protected thin spaces to between quantities and units, which depending on the PDF viewer and zoom setting might be occasionally overlooked. We have checked again the typesetting of quantities and units in the manuscript and made it more consistent at several locations.

[Figure]

Figure 1: (a) Soil types in the region around the Fendt site, based on BGR (2013), denoted in WRB classification (IUSS Working Group WRB, 2015). 'Fl./Gl.' stands for 'Fluvisols/Gleysols'. (b) Simplified land cover map (CORINE 2012 v18.5, European Environment Agency, EEA (2016)) of the same region. In both panels the location of the Fendt site is marked with a black diamond.

[Figure]

Figure 2: Statistics of NEE fluxes obtained with the EC technique during July 2016

[Figure]

Figure 3: Height $z$ versus distance travelled $d$ of 200 particles released on 6 July at 21:00 UTC at a height of 10 m as they travel back in time until $t_0 =$18:00 UTC

[Figure]

Figure 4: Same as Fig. 3, but for a release height of 100 m

[Figure]

Figure 5: Same as Fig. 3, but for particle release on 9 July at 21:00 UTC

[Figure]

Figure 6: Same as Fig. 3, but for particle release on 9 July at 21:00 UTC at a height of 100 m

[Figure]

Figure 7: Trajectories of particles released at a height of $10\,\mathrm{m}$ at 21:00 UTC on two different days as they travel backwards until $t_0 =$18:00 UTC of the same day

[revised manuscript text omitted]

---

## Author Response (AR2)

**Author's response to editor comments on amt-2019-221**

**11 February 2020**

We thank the editor for his helpful comments, which are listed below in italics, followed by our response typeset upright. Changes to the manuscript are highlighted in blue colour.

A marked-up version of the revised manuscript is attached.

**Comments**

*p1 line 3 / 19: At both locations, a better explanation is necessary. It needs to be specified that (nocturnal) stable stratification does not generally violate the assumptions for EC measurement, but only under 'calm and stable conditions' (or 'very stable conditions') without fully developed turbulence. Windy stable conditions, which can also frequently occur during nights at many sites, are not problematic for EC measurements.*

Changed to 'calm and stable conditions' and 'Stable conditions combined with low wind speeds', respectively.

*p1 line 10: change to "an area in the order of"*

Changed

*p1 line 16: Why is "methane" introduced here but not mentioned in the rest of the introduction? I suggest to leave it away and fully concentrate on CO2.*

Changed

*p2 line 12: Why do you mix/combine "Biometric approaches, including enclosure-based methods". These are two totally different approaches in my view with different advantages/disadvantages.*

We agree and changed the sentence to 'Enclosure-based methods and biometric approaches, including plant growth assessment and stock inventories, are often employed [...]'.

*p2 line 20/21: I do not agree that going from the EC scale to a larger scale is generally the better choice than going to the smaller scale (chambers). This is not supported by existing literature and also not by the present study (that uses small chambers as a reference). Instead the authors should argue that going to the larger scale is an 'alternative option' to chambers with other advantages and disadvantages. I would even suggest not to stress the potential of the NBL budget as direct alternative to nighttime EC (or chambers), but rather as (the only) method being able to measure fluxes on a larger (landscape) scales.*

It was not our intention to state that NBL budgets are better suited for validating EC measurements than enclosure-based approaches. Our argumentation is rather that (1) EC measurements are unreliable under calm and stable conditions, hence validation and correction is needed, (2) enclosure-based and biometric approaches are often used for this purpose, but due to the smaller footprint this comparison is prone to sampling biases, hence they cannot be the full answer to the problem, (3) no method is known that measures fluxes independently of the EC

technique with the same footprint as EC measurements (this would be the optimal case with no risk for sampling bias), hence (4) NBL budgets with their larger footprint are the best option to constrain EC errors better than would be possible with chambers and biometric approaches alone. To make this clearer, we revised ll. 13–23 as follows: 'The nighttime problem of EC measurements calls for error quantification and potentially correction. Ideally, this would be achieved by comparison to a method that is sensitive to fluxes in the same area, but is not based on the same assumptions as the EC technique. Unfortunately, no such method is available. Constraining the error of EC measurements must therefore rely on methods that determine fluxes on smaller and larger scales.

Enclosure-based methods and biometric approaches, including plant growth assessment and stock inventories, are often employed to obtain independent estimates for NEE [**?**, **?**, **?**]. These methods quantify the exchange of carbon on a much smaller spatial scale than EC measurements. The chambers typically used for determining soil respiration cover an area of less than one square meter, while the EC technique is sensitive to fluxes from an area of $10^4$–$10^6 \, \mathrm{m}^2$, depending on the site and on meteorological conditions [**?**]. Given these different scales, inhomogeneities in the ecosystem under study, such as spatial variability of soil properties (e.g., texture, carbon content, nitrogen content), soil environmental conditions (e.g., soil temperature and moisture) or plant community composition can lead to biases in the comparison.

In order to counteract these biases, larger-scale flux estimates should be obtained in addition to enclosure based or biometric measurements when constraining the error of EC measurements. Nocturnal boundary layer (NBL) budgets [...]'

*p2 line 25: Clarify to "Any tracer emitted from the surface into the atmosphere..."*
Changed

*p3 line 5: I suggest to write "...at low instrumental costs." The instrumentation may be low cost with this method, but the manpower for operation over certain time periods is definitely not low cost (in contrast to an EC or chamber system that is run automatically). Maybe you also want to comment on that latter issue.*

In this sentence we state that UAS have 'the potential to probe the NBL [...] at low cost'. Neither EC nor chamber systems yield vertically resolved measurements of a significant part of a the NBL, hence they are not 'competitors' in that respect. The two most common platforms to obtain such measurements are manned aircraft and tall towers. UAS-based measurements beat both of them in instrumental cost and cost of operation, so the sentence seems justified and we would prefer to leave it as it is. Taking hourly nighttime measurements over the course of several weeks with a non-automated UAS-based approach would of course be much more expensive than operating a set of automated chambers over the same time span, but such a measurement strategy is not recommended at any point in the manuscript.

*p3 line 14: Replace "creation" by "determination"; line 19: Replace accordingly "created" by "determined"*
Changed there and at several other occurences

*p3 line 16/17: The sentence "In the following ..." can be omitted.*
Changed

*p3 line 29: Change "sowed" to "sown"*
Changed

*p3 line 33: It would be good to indicate the position and extension of this grid in Fig. 1 or 2.*

We added '[...] a regular grid covering an area of 300 m by 300 m'. This grid would be hardly visible at the scale of Fig. 2. In the right panel of Fig. 1 it would still be small and partially hide the land cover, which to show is the main purpose of that panel. We also think that the soil grid is not important enough to warrant a separate figure.

*p3 line 33: "5 cm depth" occurs twice in the sentence. Omit one.*
Changed

*p7 line 30: Replace "lower than 150m" by "less than 150 m"*
Changed

*p8 Eq. 2: It is not clear what the subscript "SC" of x_SC(t) means here.*
Eq. 2 is introduced with the sentence 'Its response to step-changes in $CO_2$ dry air mole fraction can be approximated as $x_{SC}(t) =$ [...]'. 'SC' stands for step-change and is used here to be coherent with the cited reference. We do not see how we could make this clearer.

*p11 line 10: What do "l1, l2" mean here. I think they are not defined in the text.*
Indeed $l_1$ and $l_2$ are not yet introduced as spatial coordinates at this point. We changed the sentence to 'In order to evaluate the second integral in term A, $\int_0^{z_t} c(z, t_0)\, dz$, we assume horizontal and vertical homogeneity of the $CO_2$ dry air mole fraction at the time $t_0$, i.e. $x_{CO2}(t_0)$ is assumed to be constant within the spatial domain relevant for our experiments'.

*p11 line 14/15: The sentence "The sign of the daytime CO2..." is not necessary and a bit confusing at this position. I suggest to omit it.*
The sentence plays an important role in the argumentation. Ecosystems that are a strong $CO_2$ source at night often (not always) take up a lot of $CO_2$ during daytime. This finding leads to the question how horizontal homogeneity of concentration can be a weaker assumption during daytime than during nighttime, which is answered in the following two sentences in the manuscript. If nighttime emission was e.g. a hundred times smaller than daytime uptake, the argument of mixing height would be meaningless. We revised said sentence to put more emphasis on the magnitude and less emphasis on the sign: ' Daytime and nighttime $CO_2$ flux of a vegetated area are usually of the same order of magnitude, although different in sign.'

*p11 line 21: The sentence "In a stable NBL, however, little turbulent exchange takes place" is not really correct. Du to (intermittent) turbulent exchange, the CO2 emitted at the surface is mixed up to heights of e.g. 100 m as shown in Fig. 7. Probably the authors wanted to say that "little (or no) turbulent exchange takes place across the top of the NBL"?*
Changed to 'However, when a stable NBL has developed, little turbulent exchange takes place across the top of the NBL.'

*p13 line 21/22: I think it should read here "...with a Gaussian filter ..."*
Changed to 'with a Gaussian function' here and in the caption of Fig. 4 and 5.

*p14 line 10/11: I do not really see the sense of smoothing the height z, as long as the filter for the concentration profile is symmetric and a function of height (and not of number of available measurement points). Was that not the case here?*
As described in the text, the smoothing was carried out over a defined time span, translating to a fixed number of measurement points. This has the benefit of reaching a defined noise level and is computationally efficient.

*p18 line 27: The term "stationary measurements" could be misleading in the micrometeorological context (it is usually used for temporal stationarity of measurements). Better use e.g. "tower-based measurements"*
Changed here and at two other occurences

*Figure 9: It should be clarified in the figure caption that the HPB data was a measurement at a single height above the NBL, which is assumed representative for the entire (daytime) profile. In order to distinguish this data from the fully measured COPAP profiles, I suggest to plot the HPB concentration as a dashed line.*
We followed the suggestion by plotting the $CO_2$ dry air mole fraction measured at HPB as a dashed line and by changing the caption to '[...] by the ICOS station HPB (a point measurement

460 m above ground level at Fendt as described in Section 2.2, representing the $CO_2$ dry air mole fraction in the residual layer) and by the on-site 9 m mast [...]'

*Figure 9: The figure caption statement "for each 1-minute sampling period of the mast" is not fully clear. Did you mean "...during the individual UAS flight" ?*

This was explained in the text, we added the same description to the caption.

*p19 line 10/11: The statement "that the stable NBL retains most of the surface fluxes." is sloppy and needs rephrasing. E.g. "...retains most of the surface emitted CO2."*

Changed

*Table 2: List also the duration of the flights in this table.*

The flight duration is short in comparison to the time since $t_0$, so the information seems not relevant at this point and would rather distract the reader. We agree, however, that the exact flight times should be part of the manuscript. For this purpose we introduce another table (Table 2 in the revised manuscript):

Table 1: Take-off and landing times (UTC)

| Flight | Date | Take-off | Landing | Duration |
|--------|------|----------|---------|----------|
| | year-month-day | hour:minute | hour:minute | |
| #4 | 2016-07-06 | 18:04 | 18:14 | 10 min 50 s |
| #5 | 2016-07-06 | 19:05 | 19:16 | 10 min 58 s |
| #6 | 2016-07-06 | 21:13 | 21:23 | 10 min 13 s |
| #7 | 2016-07-06 | 22:09 | 22:19 | 09 min 20 s |
| #8 | 2016-07-06 | 23:06 | 23:16 | 09 min 43 s |
| #9 | 2016-07-07 | 00:18 | 00:28 | 09 min 23 s |
| #10 | 2016-07-07 | 01:11 | 01:20 | 09 min 18 s |
| #19 | 2016-07-09 | 20:01 | 20:11 | 09 min 58 s |
| #20 | 2016-07-09 | 21:02 | 21:13 | 11 min 31 s |
| #21 | 2016-07-09 | 22:43 | 22:55 | 11 min 23 s |
| #22 | 2016-07-09 | 23:39 | 23:49 | 10 min 34 s |
| #23 | 2016-07-10 | 00:31 | 00:42 | 10 min 32 s |

To avoid confusion of the careful reader, we add to Sect. 4.3: 'The times given in Fig. 6 and 7 are the midtimes of the flights rounded to full 10 minutes for readability. The exact times of take-off and landing are provided in Table 2.' For the same reason, we add to the caption of Table 3 (formerly Table 2): 'The end time is specified as the midtime of the portion of the flight used for determination of the NBL budget.'

*p30 line 13: Better write: "Firstly, this would support ..."*

Changed

[revised manuscript text omitted]